# BAYESIAN EXPERIMENTAL DESIGN VIA CONTRASTIVE DIFFUSIONS

**Jacopo Iollo**[1,2,3]       **Christophe Heinkelé**[2]       **Pierre Alliez**[3]       **Florence Forbes**[1]

1: Université Grenoble Alpes, Inria, CNRS, G-INP, France, `name.surname@inria.fr`

2: Cerema, Endsum-Strasbourg, France, `christophe.heinkele@cerema.fr`

3: Université Côte d'Azur, Inria, France, `name.surname@inria.fr`

## ABSTRACT

Bayesian Optimal Experimental Design (BOED) is a powerful tool to reduce the cost of running a sequence of experiments. When based on the Expected Information Gain (EIG), design optimization corresponds to the maximization of some intractable expected *contrast* between prior and posterior distributions. Scaling this maximization to high dimensional and complex settings has been an issue due to BOED inherent computational complexity. In this work, we introduce a *pooled posterior* distribution with cost-effective sampling properties and provide a tractable access to the EIG contrast maximization via a new EIG gradient expression. Diffusion-based samplers are used to compute the dynamics of the pooled posterior and ideas from bi-level optimization are leveraged to derive an efficient joint sampling-optimization loop. The resulting efficiency gain allows to extend BOED to the well-tested generative capabilities of diffusion models. By incorporating generative models into the BOED framework, we expand its scope and its use in scenarios that were previously impractical. Numerical experiments and comparison with state-of-the-art methods show the potential of the approach.

## 1 INTRODUCTION

Designing optimal experiments can be critical in numerous applied contexts where experiments are constrained in terms of resources or more generally costly and limited. In this work, design is assumed to be characterized by some continuous parameters $\boldsymbol{\xi} \in \mathcal{E} \subset \mathbb{R}^d$, which refers to the experimental part, such as the choice of a measurement location, that can be controlled to optimize the experimental outcome. We consider a Bayesian setting in which the parameters of interest is $\boldsymbol{\theta} \in \boldsymbol{\Theta} \subset \mathbb{R}^m$ and design is optimized to maximize the information gain on $\boldsymbol{\theta}$. Bayesian optimal experimental design (BOED) is not a new topic in statistics, see *e.g.* Chaloner and Verdinelli (1995); Sebastiani and Wynn (2000); Amzal et al. (2006) but has recently gained new interest with the use of machine learning techniques, see Rainforth et al. (2024); Huan et al. (2024) for recent reviews. The most common approach consists of maximizing the so-called expected information gain (EIG), which is a mutual information criterion that accounts for information via the Shannon's entropy. Let $p(\boldsymbol{\theta})$ denote a prior probability distribution and $p(\boldsymbol{y}|\boldsymbol{\theta}, \boldsymbol{\xi})$ a likelihood defining the observation $\boldsymbol{y} \in \mathcal{Y}$ generating process. The prior is assumed to be independent on $\boldsymbol{\xi}$ and $p(\boldsymbol{y}|\boldsymbol{\theta}, \boldsymbol{\xi})$ available in closed-form. To our knowledge, all previous BOED approaches also assume that the prior is available in closed-form, a setting that we refer to as **density-based BOED**. In this work, by making BOED more computationally efficient, we open the first access to diffusion-based generative models and introduce **data-based BOED** when the prior is only available through samples. This broadens the scope of problems that can be tackled to a wide range of inverse problems (Daras et al., 2024).

The EIG, denoted below by $I$, admits several equivalent expressions, see *e.g.* Foster et al. (2019). It can be written as the expected loss in entropy when accounting for an observation $\boldsymbol{y}$ at $\boldsymbol{\xi}$ (eq. (1)) or as a mutual information (MI) or expected Kullback-Leibler (KL) divergence (eq. (2)). Denoting $p_{\boldsymbol{\xi}}(\boldsymbol{\theta}, \boldsymbol{y}) = p(\boldsymbol{\theta}, \boldsymbol{y}|\boldsymbol{\xi})$ the joint distribution of $(\boldsymbol{\theta}, \boldsymbol{Y})$ and using $p(\boldsymbol{\theta}, \boldsymbol{y}|\boldsymbol{\xi}) = p(\boldsymbol{\theta}|\boldsymbol{y}, \boldsymbol{\xi})p(\boldsymbol{y}|\boldsymbol{\xi}) = p(\boldsymbol{y}|\boldsymbol{\theta}, \boldsymbol{\xi})p(\boldsymbol{\theta})$, it comes,

$$I(\boldsymbol{\xi}) = \mathbb{E}_{p(\boldsymbol{y}|\boldsymbol{\xi})}[\mathrm{H}(p(\boldsymbol{\theta})) - \mathrm{H}(p(\boldsymbol{\theta}|\boldsymbol{Y}, \boldsymbol{\xi}))] \tag{1}$$

$$= \mathbb{E}_{p(\boldsymbol{y}|\boldsymbol{\xi})}\left[\mathrm{KL}(p(\boldsymbol{\theta}|\boldsymbol{Y}, \boldsymbol{\xi}), p(\boldsymbol{\theta}))\right] = \mathrm{MI}(p_{\boldsymbol{\xi}}), \tag{2}$$

where random variables are indicated with uppercase letters, $\mathbb{E}_{p(\cdot)}[\cdot]$ or $\mathbb{E}_p[\cdot]$ denotes the expectation with respect to $p$ and $\mathrm{H}(p(\boldsymbol{\theta})) = -\mathbb{E}_{p(\boldsymbol{\theta})}[\log p(\boldsymbol{\theta})]$ is the entropy of $p$. The joint distribution $p_{\boldsymbol{\xi}}$ completely determines all other distributions, marginal (prior) and conditional (posterior) distributions, so that the mutual information, which is the KL between the joint and the product of its marginal distributions, can be written as a function of $p_{\boldsymbol{\xi}} \in \mathcal{P}(\boldsymbol{\Theta} \times \mathcal{Y})$ only. In the following $\mathcal{P}(\boldsymbol{\Theta} \times \mathcal{Y})$, resp. $\mathcal{P}(\boldsymbol{\Theta})$, resp. $\mathcal{P}(\mathcal{Y})$, denotes the set of probability measures on $\boldsymbol{\Theta} \times \mathcal{Y}$, resp. $\boldsymbol{\Theta}$, resp. $\mathcal{Y}$.

In BOED, we look for $\boldsymbol{\xi}^*$ satisfying

$$\boldsymbol{\xi}^* \in \arg \max_{\boldsymbol{\xi} \in \mathbb{R}^d} I(\boldsymbol{\xi}) = \arg \max_{\boldsymbol{\xi} \in \mathbb{R}^d} \mathrm{MI}(p_{\boldsymbol{\xi}}) \ . \tag{3}$$

The above optimization is usually referred to as static design optimization. The main challenge in EIG-based BOED is that both the EIG and its gradient with respect to $\boldsymbol{\xi}$ are doubly intractable. Their respective expressions involve an expectation of an intractable integrand over a posterior distribution which is itself not straightforward to sample from. The posterior distribution is generally only accessible through an iterative algorithm providing approximate samples. In practice, the inference problem is further complicated as design optimization is considered in a sequential context, in which a series of experiments is planned sequentially and each successive design has to be accounted for. In order to remove the integrand intractability issue, solutions have been proposed which optimize an EIG lower bound (Foster et al., 2019). This lower bound can be expressed as an expectation of a tractable integrand and becomes tight with increased simulation budgets. The remaining posterior sampling issue has then been solved in different ways. A set of approaches consists of approximating the problematic posterior distribution, either with variational techniques (Foster et al., 2019) or with efficient sequential Monte Carlo (SMC) sampling (Iollo et al., 2024; Drovandi et al., 2013). Other approaches avoid posterior estimation, using reinforcement learning (RL) and off-line policy learning to bypass the need for sampling (Foster et al., 2021; Ivanova et al., 2021; Blau et al., 2022). However, some studies have shown that estimating the posterior was beneficial, *e.g.* Iollo et al. (2024) and Ivanova et al. (2024), which improves on Foster et al. (2021) by introducing posterior estimation steps in order to refine the learned policy. In addition, one should keep in mind that posterior inference is central in BOED as the ultimate goal is not design per se but to gain information on the parameter of interest. This is challenging especially in a sequential context. Previous attempts that provide both candidate design and estimates of the posterior distribution, such as Foster et al. (2019); Iollo et al. (2024), are thus essentially in 2 alternating stages, approximate design optimization being dependent on approximate posterior sampling and vice-versa.

In this work, we propose a novel 1-stage approach which leverages a sampling-as-optimization setting (Korba and Salim, 2022; Marion et al., 2025) where sampling is seen as an optimization task over the space of probability distributions. We introduce a new EIG gradient expression (Section 3), which highlights the EIG gradient as a function of both the design and some sampling outcome. This fits into a bi-level optimization framework adapted to BOED in Section 4. So doing, at each step, both an estimation of the optimal design and samples from the current posterior distribution can be provided in a single loop described in Section 5. It results an efficient procedure that can handle both traditional density-based samplers and data-based samplers such as provided by the highly successful diffusion-based generative models. The resulting efficiency gain enables BOED applications at significantly larger scales than previously feasible, including inpainting problems ranging from computer vision to protein engineering and MRI (Quan et al., 2024; Yang et al., 2019; Aali et al., 2023). Figure 1 is an illustration on a $28 \times 28$ image $\boldsymbol{\theta}$ reconstruction problem from $7 \times 7$ sub-images centered at locations $\boldsymbol{\xi}$ to be selected, details in Section 6. For simpler notation, we first present our approach in the static design case. Adaptation to the sequential case is specified in Section 6 and all numerical examples are in the sequential setting.

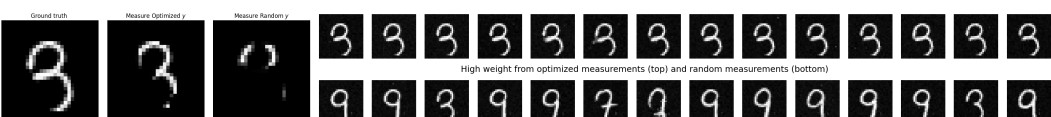

Figure 1: $28 \times 28$ Image $\boldsymbol{\theta}$ (1st column) reconstruction from seven $7 \times 7$ sub-images $\boldsymbol{y} = \boldsymbol{A}_{\boldsymbol{\xi}}\boldsymbol{\theta} + \boldsymbol{\eta}$ centered at seven central pixels $\boldsymbol{\xi}$ (designs) selected sequentially. Optimized vs. random designs: measured outcome $\boldsymbol{y}$ (2nd vs. 3rd column) and parameter $\boldsymbol{\theta}$ estimates (reconstruction) with highest weights (upper vs. lower sub-row).

## 2 RELATED WORK

We focus on gradient-based BOED for continuous problems. Applying a first-order method to solve (3) requires computing gradients of the EIG $I$, which are no more tractable than $I$ itself. Gradient-based BOED is generally based on stochastic gradient-type algorithms (see Section 4.3.2. in Huan et al. (2024)). This requires in principle unbiased gradient estimators, although stochastic approximation solutions using biased oracles have also been investigated, see *e.g.* Demidovich et al. (2023); Liu and Tajbakhsh (2024). To meet this requirement, most stochastic gradient-based approaches start from an EIG lower bound that yields tractable unbiased gradient estimators. More specifically, EIG lower bounds have usually the advantage to remove the nested expectation issue, see *e.g.* Foster et al. (2019). In contrast, very few approaches focus on direct EIG gradient estimators. To our knowledge, this is only the case in Goda et al. (2022) and Ao and Li (2024). Goda et al. (2022) propose an unbiased estimator of the EIG gradient using a randomized version of a multilevel nested Monte Carlo (MLMC) estimator from Rhee and Glynn (2015). A different estimator is proposed by Ao and Li (2024), who use MCMC samplers leading to biased estimators, for which the authors show empirically that the bias could be made negligible. In this work, we first show, in Section 3, that their two apparently different solutions actually only differ in the way the intractable posterior distribution is approximated. We then propose a third way to compute EIG gradients that is more computationally efficient and scales better to larger data volumes and sequential design contexts. This new expression makes use of a distribution that we introduce and name the *pooled posterior* distribution. This latter distribution has interesting sampling features that allow us to leverage score-based sampling techniques and connect to the so-called *implicit diffusion* framework of Marion et al. (2025). Our single loop procedure in Section 5 is inspired by Marion et al. (2025) and other recent developments in bi-level optimization (Yang et al., 2021; Dagréou et al., 2022; Hong et al., 2023). However, these latter settings do not cover doubly intractable objectives such as the EIG, which requires both appropriate gradient estimators and sampling operators, see our Sections 3 and 4. In BOED, efficient single loop procedures have been proposed by Foster et al. (2020) but they rely heavily on variational approximations, which may limit accuracy in scenarios with complex posterior distributions.

## 3 POOLED-POSTERIOR ESTIMATION OF THE EIG GRADIENT

Efficient EIG gradient estimators are central for accurate scalable BOED. Gradients derived from the reparameterization trick are often preferred, over the ones obtained with score-based techniques, as they have been reported to exhibit lower variance (Xu et al., 2019).

**EIG gradient via a reparametrization trick.** Assuming $p(\boldsymbol{y}|\boldsymbol{\theta}, \boldsymbol{\xi})$ is such that $\boldsymbol{Y}$ can be rewritten as $\boldsymbol{Y} = T_{\boldsymbol{\xi}, \boldsymbol{\theta}}(\boldsymbol{U})$ with $T_{\boldsymbol{\xi}, \boldsymbol{\theta}}$ invertible so that $\boldsymbol{U} = T_{\boldsymbol{\xi}, \boldsymbol{\theta}}^{-1}(\boldsymbol{Y})$ and $\boldsymbol{U}$ is a random variable independent on $\boldsymbol{\theta}$ and $\boldsymbol{\xi}$ with a tractable distribution $p_U(\boldsymbol{U})$. The existence of $T_{\boldsymbol{\xi}, \boldsymbol{\theta}}$ is straightforward if the direct model corresponds to an additive Gaussian noise as the transformation is then linear in $\boldsymbol{U}$. Results exist to guarantee the existence of such a transformation in more general situations (Papamakarios et al., 2021). Using this change of variable, two expressions of the EIG gradient, (5) and (6) below, can be derived. Detailed steps are given in Appendix A. With $p_{\boldsymbol{\xi}}$ denoting the joint distribution $p(\boldsymbol{\theta}, \boldsymbol{y}|\boldsymbol{\xi})$, $g$ a quantity related to the score $g(\boldsymbol{\xi}, \boldsymbol{y}, \boldsymbol{\theta}, \boldsymbol{\theta}') = \nabla_{\boldsymbol{\xi}} \log p(T_{\boldsymbol{\xi}, \boldsymbol{\theta}}(\boldsymbol{u})|\boldsymbol{\theta}', \boldsymbol{\xi})_{|\boldsymbol{u} = T_{\boldsymbol{\xi}, \boldsymbol{\theta}}^{-1}(\boldsymbol{y})}$ and denoting $h(\boldsymbol{\xi}, \boldsymbol{y}, \boldsymbol{\theta}, \boldsymbol{\theta}') = \nabla_{\boldsymbol{\xi}} p(T_{\boldsymbol{\xi}, \boldsymbol{\theta}}(\boldsymbol{u})|\boldsymbol{\theta}', \boldsymbol{\xi})_{|\boldsymbol{u} = T_{\boldsymbol{\xi}, \boldsymbol{\theta}}^{-1}(\boldsymbol{y})}$, a first expression is

$$\nabla_{\boldsymbol{\xi}} I(\boldsymbol{\xi}) = \mathbb{E}_{p_{\boldsymbol{\xi}}} \left[ g(\boldsymbol{\xi}, \boldsymbol{Y}, \boldsymbol{\theta}, \boldsymbol{\theta}) - \frac{\mathbb{E}_{p(\boldsymbol{\theta}')} \left[ h(\boldsymbol{\xi}, \boldsymbol{Y}, \boldsymbol{\theta}, \boldsymbol{\theta}') \right]}{\mathbb{E}_{p(\boldsymbol{\theta}')} \left[ p(\boldsymbol{Y}|\boldsymbol{\theta}', \boldsymbol{\xi}) \right]} \right]. \quad (4)$$

Considering importance sampling formulations for the second term of (4), with an importance distribution $q \in \mathcal{P}(\boldsymbol{\Theta})$, potentially depending on $\boldsymbol{y}$, $\boldsymbol{\theta}$ and $\boldsymbol{\xi}$, further leads to

$$\nabla_{\boldsymbol{\xi}} I(\boldsymbol{\xi}) = \mathbb{E}_{p_{\boldsymbol{\xi}}} \left[ g(\boldsymbol{\xi}, \boldsymbol{Y}, \boldsymbol{\theta}, \boldsymbol{\theta}) - \frac{\mathbb{E}_{q(\boldsymbol{\theta}'|\boldsymbol{Y}, \boldsymbol{\theta}, \boldsymbol{\xi})} \left[ \frac{p(\boldsymbol{\theta}')}{q(\boldsymbol{\theta}'|\boldsymbol{Y}, \boldsymbol{\theta}, \boldsymbol{\xi})} h(\boldsymbol{\xi}, \boldsymbol{Y}, \boldsymbol{\theta}, \boldsymbol{\theta}') \right]}{\mathbb{E}_{q(\boldsymbol{\theta}'|\boldsymbol{Y}, \boldsymbol{\theta}, \boldsymbol{\xi})} \left[ \frac{p(\boldsymbol{\theta}')}{q(\boldsymbol{\theta}'|\boldsymbol{Y}, \boldsymbol{\theta}, \boldsymbol{\xi})} p(\boldsymbol{Y}|\boldsymbol{\theta}', \boldsymbol{\xi}) \right]} \right]. \quad (5)$$

In Goda et al. (2022), this latter expression is used in a randomized MLMC procedure with $q$ set to a Laplace approximation of the posterior distribution, without justification for this specific choice of $q$.

It results an estimator which is not unbiased but can be de-biased following Rhee and Glynn (2015). Alternatively, a second expression of the EIG gradient is the starting point of Ao and Li (2024),

$$\nabla_{\boldsymbol{\xi}} I(\boldsymbol{\xi}) = \mathbb{E}_{p_{\boldsymbol{\xi}}} \left[ g(\boldsymbol{\xi}, \boldsymbol{Y}, \boldsymbol{\theta}, \boldsymbol{\theta}) - \mathbb{E}_{p(\boldsymbol{\theta}'|\boldsymbol{Y}, \boldsymbol{\xi})} \left[ g(\boldsymbol{\xi}, \boldsymbol{Y}, \boldsymbol{\theta}, \boldsymbol{\theta}') \right] \right] \ . \tag{6}$$

It follows a nested Monte Carlo estimator (30) given in Appendix A, using samples $\{(\boldsymbol{y}_i, \boldsymbol{\theta}_i)\}_{i=1:N}$ from the joint $p_{\boldsymbol{\xi}}$ and for each $\boldsymbol{y}_i$, samples $\{\boldsymbol{\theta}'_{i,j}\}_{j=1:M}$ from an MCMC procedure approximating the intractable posterior $p(\boldsymbol{\theta}'|\boldsymbol{y}_i, \boldsymbol{\xi})$. Interestingly, expression (6) can also be recovered by setting the importance proposal $q(\boldsymbol{\theta}'|\boldsymbol{y}, \boldsymbol{\theta}, \boldsymbol{\xi})$ to $p(\boldsymbol{\theta}'|\boldsymbol{y}, \boldsymbol{\xi})$ in (5), which provides a clear justification of why the choice of $q$ made in Goda et al. (2022) is relevant. Approaches by Goda et al. (2022) and Ao and Li (2024) thus mainly differ in their choice of approximations for the posterior distribution. Using a Laplace approximation as in Goda et al. (2022) is relevant only if the posterior is unimodal, which may not be the case in practice. The MCMC version of Ao and Li (2024) is then potentially more general but also more costly as it requires running $N$ times a MCMC sampler, targeting each time a different posterior $p(\boldsymbol{\theta}|\boldsymbol{y}_i, \boldsymbol{\xi})$. In the next paragraph, we introduce the *pooled posterior* distribution and derive another, more computationally efficient, gradient expression.

**Importance sampling EIG gradient estimator with a** *pooled posterior* **proposal.** In their work, Ao and Li (2024) consider only static design, which hides the fact that for more realistic sequential design contexts, their solution is not tractable due to its computational complexity. Their solution faces the standard issue of nested estimation (Rainforth et al., 2018). To avoid this issue we propose to use an importance sampling expression for the second term in (6), which has the advantage to move the dependence on $\boldsymbol{y}$ (and $\boldsymbol{\theta}$) from the sampling part to the integrand part. We consider a proposal distribution $q \in \mathcal{P}(\boldsymbol{\theta})$ that does not depend on $\boldsymbol{Y}$ nor $\boldsymbol{\theta}$. It comes,

$$\nabla_{\boldsymbol{\xi}} I(\boldsymbol{\xi}) = \mathbb{E}_{p_{\boldsymbol{\xi}}} \left[ g(\boldsymbol{\xi}, \boldsymbol{Y}, \boldsymbol{\theta}, \boldsymbol{\theta}) - \mathbb{E}_{q(\boldsymbol{\theta}'|\boldsymbol{\xi})} \left[ \frac{p(\boldsymbol{\theta}'|\boldsymbol{Y}, \boldsymbol{\xi})}{q(\boldsymbol{\theta}'|\boldsymbol{\xi})} \, g(\boldsymbol{\xi}, \boldsymbol{Y}, \boldsymbol{\theta}, \boldsymbol{\theta}') \right] \right], \tag{7}$$

and an approximate gradient can be obtained as

$$\frac{1}{N} \sum_{i=1}^{N} \left[ g(\boldsymbol{\xi}, \boldsymbol{y}_i, \boldsymbol{\theta}_i, \boldsymbol{\theta}_i) - \mathbb{E}_{q(\boldsymbol{\theta}'|\boldsymbol{\xi})} \left[ \frac{p(\boldsymbol{\theta}'|\boldsymbol{y}_i, \boldsymbol{\xi})}{q(\boldsymbol{\theta}'|\boldsymbol{\xi})} \, g(\boldsymbol{\xi}, \boldsymbol{y}_i, \boldsymbol{\theta}_i, \boldsymbol{\theta}') \right] \right] \ . \tag{8}$$

The second term in (8) still requires $N$ importance sampling approximations whose quality depends on the choice of the proposal distribution $q$. The ideal proposal $q$ is easy to simulate, with computable weights at least up to a constant, and so that $q$ and the multiple target distributions $p(\cdot|\boldsymbol{y}_i, \boldsymbol{\xi})$ are not too far apart. Given $N$ samples $\{(\boldsymbol{\theta}_i, \boldsymbol{y}_i)\}_{i=1:N}$ from $p_{\boldsymbol{\xi}}$, we propose thus to take $q = q_{\boldsymbol{\xi}, N}$ where $q_{\boldsymbol{\xi}, N}$ is the following logarithmic pooling or geometric mixture, with $\sum_{i=1}^{N} \nu_i = 1$,

$$q_{\boldsymbol{\xi}, N}(\boldsymbol{\theta}) \propto \prod_{i=1}^{N} p(\boldsymbol{\theta}|\boldsymbol{y}_i, \boldsymbol{\xi})^{\nu_i} \propto p(\boldsymbol{\theta}) \prod_{i=1}^{N} p(\boldsymbol{y}_i|\boldsymbol{\theta}, \boldsymbol{\xi})^{\nu_i} \ . \tag{9}$$

We refer to $q_{\boldsymbol{\xi}, N}$ as the *pooled posterior* distribution, defined in a more general way in (13). It allows to assess the effect of a candidate design $\boldsymbol{\xi}$ on samples from the prior. It differs from a standard posterior as no real data $\boldsymbol{y}$ obtained by running the experiment $\boldsymbol{\xi}$ is available during the optimization. We only have access to samples $\{(\boldsymbol{\theta}_i, \boldsymbol{y}_i)\}_{i=1:N}$ from the joint $p_{\boldsymbol{\xi}}$. The pooled posterior can be seen as a distribution that takes into account all possible outcomes of a candidate experiment $\boldsymbol{\xi}$ given the samples $\{\boldsymbol{\theta}_i\}_{i=1:N}$ from the prior. This choice of $q_{\boldsymbol{\xi}, N}$ is justified in Appendix B, using Lemma 2, proved therein. Lemma 2 shows that, for $\sum_{i=1}^{N} \nu_i = 1$, $q_{\boldsymbol{\xi}, N}$ is the distribution $q$ that minimizes the weighted sum of the KLs against each posterior $p(\boldsymbol{\theta}|\boldsymbol{y}_i, \boldsymbol{\xi})$, *i.e.* $\sum_{i=1}^{N} \nu_i \text{KL}(q, p(\boldsymbol{\theta}|\boldsymbol{y}_i, \boldsymbol{\xi}))$, leading to an efficient importance sampling proposal. It follows our new gradient estimator,

$$\nabla_{\boldsymbol{\xi}} I(\boldsymbol{\xi}) \approx \frac{1}{N} \sum_{i=1}^{N} \left[ g(\boldsymbol{\xi}, \boldsymbol{y}_i, \boldsymbol{\theta}_i, \boldsymbol{\theta}_i) - \frac{1}{M} \sum_{j=1}^{M} w_{i,j} \, g(\boldsymbol{\xi}, \boldsymbol{y}_i, \boldsymbol{\theta}_i, \boldsymbol{\theta}'_j) \right], \tag{10}$$

where $\{(\boldsymbol{\theta}_i, \boldsymbol{y}_i)\}_{i=1:N}$ follow $p_{\boldsymbol{\xi}}$, $\{\boldsymbol{\theta}'_i\}_{j=1:M}$ follow $q_{\boldsymbol{\xi}, N}$ and $w_{i,j} = \frac{p(\boldsymbol{\theta}'_j|\boldsymbol{y}_i, \boldsymbol{\xi})}{q_{\boldsymbol{\xi}, N}(\boldsymbol{\theta}'_j)}$ denotes the importance sampling weight. When this fraction can only be evaluated up to a constant, we consider

self normalized importance sampling (SNIS) using $\tilde{p}$, $\tilde{q}_{\boldsymbol{\xi},N}$ the unnormalized versions of $p$ and $q_{\boldsymbol{\xi},N}$,

$$\tilde{w}_{i,j} = \frac{\tilde{p}(\boldsymbol{\theta}'_j|\boldsymbol{y}_i,\boldsymbol{\xi})}{\tilde{q}_{\boldsymbol{\xi},N}(\boldsymbol{\theta}'_j)} = \frac{p(\boldsymbol{y}_i|\boldsymbol{\theta}'_j,\boldsymbol{\xi})}{\prod_{\ell=1}^{N} p(\boldsymbol{y}_\ell|\boldsymbol{\theta}'_j,\boldsymbol{\xi})^{\nu_\ell}} \quad \text{and} \quad w_{i,j} = \frac{\tilde{w}_{i,j}}{\sum_{j=1}^{M} \tilde{w}_{i,j}} \ . \tag{11}$$

Although with a reduced computational cost, computing gradients with (10) still requires an iterative sampling algorithm ideally run for a large number of iterations to reach satisfying approximations of the joint $p_{\boldsymbol{\xi}}$ and the pooled posterior $q_{\boldsymbol{\xi},N}$. In static design, sampling from the joint is not generally difficult as the prior and the likelihood are assumed available but this becomes problematic in sequential design, as further detailed in Section 6.1. Sequential design is the setting to be kept in mind in this paper and in practice, the exact distributions are rarely reached. To assess the impact on gradient approximations, it is convenient to introduce, as in Marion et al. (2025), gradient *operators*. In the next section, we show how to adapt the formalism of Marion et al. (2025) to our BOED task.

## 4 EIG OPTIMIZATION THROUGH SAMPLING

To maximize the EIG using its gradient estimator (10), samples are needed from both the joint distribution $p_{\boldsymbol{\xi}}$ and the pooled posterior proposal $q_{\boldsymbol{\xi},N}$. If handled naively, it results a computationally expensive nested sampling-optimization loop where new samples from both distributions need to be generated at every update of the design parameter $\xi$. To derive more efficient procedures, we propose to adapt to BOED the framework of Marion et al. (2025) that integrates sampling and optimization into a single bi-level optimization loop. To do so, the EIG gradient $\nabla_{\xi}I(\xi)$ has first to be expressed as a function $\Gamma$ of three key components: the joint distribution $p_{\xi}$, the proposal distribution $q$, and the design parameter $\xi$ itself. Our choice of the pooled posterior as proposal distribution $q$ is then justified for its interesting sampling properties and the concept of sampling operator of Marion et al. (2025) is generalized to efficiently generate the samples needed to estimate our EIG gradient via $\Gamma$.

**Estimation of gradients through sampling.** Denote by $\Gamma$ a function from $\mathcal{P}(\boldsymbol{\Theta}\times\mathcal{Y})\times\mathcal{P}(\boldsymbol{\Theta})\times\mathbb{R}^d$ to $\mathbb{R}^d$, defined as,

$$\Gamma(p,q,\boldsymbol{\xi}) = \mathbb{E}_p\left[g(\boldsymbol{\xi},\boldsymbol{Y},\boldsymbol{\theta},\boldsymbol{\theta}) - \mathbb{E}_q\left[\frac{p(\boldsymbol{\theta}'|\boldsymbol{Y})}{q(\boldsymbol{\theta}')}g(\boldsymbol{\xi},\boldsymbol{Y},\boldsymbol{\theta},\boldsymbol{\theta}')\right]\right] \ . \tag{12}$$

Expression (7) shows that $\nabla_{\boldsymbol{\xi}}I(\boldsymbol{\xi}) = \Gamma(p_{\boldsymbol{\xi}},q,\boldsymbol{\xi})$, where $q$ is a distribution $q(\boldsymbol{\theta}'|\boldsymbol{\xi})$ on $\boldsymbol{\theta}'$ possibly depending on $\boldsymbol{\xi}$. The gradient estimator (10) corresponds then to $\nabla_{\boldsymbol{\xi}}I(\boldsymbol{\xi}) \approx \Gamma(\hat{p}_{\boldsymbol{\xi}},\hat{q}_{\boldsymbol{\xi},N},\boldsymbol{\xi})$, where $\hat{p}_{\boldsymbol{\xi}} = \sum_{i=1}^{N}\delta_{(\boldsymbol{\theta}_i,\boldsymbol{y}_i)}$ and $\hat{q}_{\boldsymbol{\xi},N} = \sum_{j=1}^{M}\delta_{\boldsymbol{\theta}'_j}$. In general, sampling from $p_{\boldsymbol{\xi}}$, or its sequential counterpart, and $q_{\boldsymbol{\xi},N}$ is challenging and only possible through an iterative procedure. However, an interesting feature of our pooled posterior is that it does not add additional sampling difficulties.

**Pooled posterior distribution.** More generally (details in Appendix B), we define,

$$q_{\boldsymbol{\xi},\rho}(\boldsymbol{\theta}) \propto \exp\left(\mathbb{E}_\rho[\log p(\boldsymbol{\theta}|\boldsymbol{Y},\boldsymbol{\xi})]\right) \tag{13}$$

where $\rho$ is a measure on $\mathcal{Y}$. When $\rho(\boldsymbol{y}) = \sum_{i=1}^{N}\nu_i\delta_{\boldsymbol{y}_i}(\boldsymbol{y})$ with $\sum_{i=1}^{N}\nu_i = 1$, we recover $q_{\boldsymbol{\xi},\rho}(\boldsymbol{\theta}) = q_{\boldsymbol{\xi},N}(\boldsymbol{\theta})$ in (9). The special structure of the pooled posterior allows to sample from it using the same algorithmic structure to sample from a single posterior $p(\boldsymbol{\theta}|\boldsymbol{y},\boldsymbol{\xi})$. Indeed, the score of $q_{\boldsymbol{\xi},\rho}(\boldsymbol{\theta})$ is linked to the posterior score,

$$\nabla_{\boldsymbol{\theta}}\log q_{\boldsymbol{\xi},\rho}(\boldsymbol{\theta}) = \mathbb{E}_\rho[\nabla_{\boldsymbol{\theta}}\log p(\boldsymbol{\theta}|\boldsymbol{Y},\boldsymbol{\xi})] \ , \tag{14}$$

which for $q_{\boldsymbol{\xi},N}$ simplifies into $\sum_{i=1}^{N}\nu_i\nabla_{\boldsymbol{\theta}}\log p(\boldsymbol{\theta}|\boldsymbol{y}_i,\boldsymbol{\xi})$. In practice, we consider the operation of sampling as the output of a stochastic process iterating a so-called *sampling operator*.

**Iterative sampling operators.** Iterative sampling operators, as introduced in Marion et al. (2025), are mappings to a space of probabilities. In our BOED setting, we consider two such operators. The first one is defined, for each $\boldsymbol{\xi}$, through a sequence over $s$ of functions from $\mathcal{P}(\boldsymbol{\Theta}\times\mathcal{Y})$ to $\mathcal{P}(\boldsymbol{\Theta}\times\mathcal{Y})$ and denoted by $\Sigma_s^{\boldsymbol{Y},\boldsymbol{\theta}}(p,\boldsymbol{\xi})$. Sampling is defined as the outcome in the limit $s\to\infty$ or for some finite $s = S$ of the following process starting from $p^{(0)}\in\mathcal{P}(\boldsymbol{\Theta}\times\mathcal{Y})$ and iterating

$$p^{(s+1)} = \Sigma_s^{\boldsymbol{Y},\boldsymbol{\theta}}(p^{(s)},\boldsymbol{\xi}) \ , \tag{15}$$

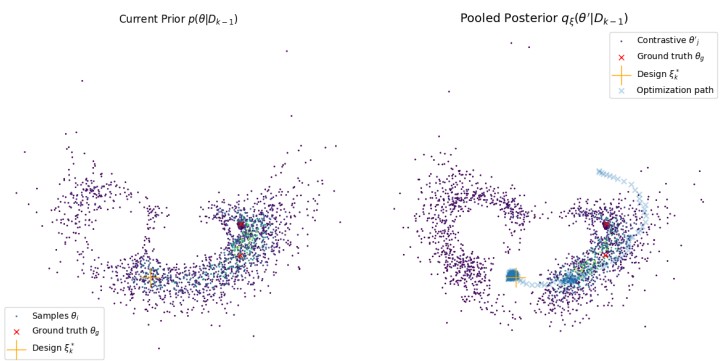

Figure 2: Source localisation example. Prior (left) and pooled posterior (right) samples at experiment $k$. Final $\boldsymbol{\xi}_k^*$ (orange cross) at the end of the optimization sequence $\boldsymbol{\xi}_0, \cdot, \boldsymbol{\xi}_T$ (blue crosses). This optimization "contrasts" the two distributions by making the pooled posterior "as different as possible" from the prior.

where $p^{(s)}$ can be explicit, *e.g.* $p^{(s)}$ is a Gaussian distribution with parameters depending on $s$, or represented by a random variable $\boldsymbol{X}_s \sim p^{(s)}$. For example, in density-based BOED, for $\boldsymbol{X}_s = (\boldsymbol{Y}_s, \boldsymbol{\theta}_s)$, we consider the Euler discretization of the Langevin diffusion converging to $p_{\boldsymbol{\xi}}$

$$
\begin{aligned}
\boldsymbol{y}^{(s+1)} &= \boldsymbol{y}^{(s)} - \gamma_s \nabla_{\boldsymbol{y}} V(\boldsymbol{y}^{(s)}, \boldsymbol{\theta}^{(s)}, \boldsymbol{\xi}) + \sqrt{2\gamma_s} \boldsymbol{B}_{\boldsymbol{y},s}, \\
\boldsymbol{\theta}^{(s+1)} &= \boldsymbol{\theta}^{(s)} - \gamma_s \nabla_{\boldsymbol{\theta}} V(\boldsymbol{y}^{(s)}, \boldsymbol{\theta}^{(s)}, \boldsymbol{\xi}) + \sqrt{2\gamma_s} \boldsymbol{B}_{\boldsymbol{\theta},s}.
\end{aligned}
\tag{16}
$$

where $\boldsymbol{B}_{\boldsymbol{y},s}$ and $\boldsymbol{B}_{\boldsymbol{\theta},s}$ are realizations of independent standard Gaussian variables, $\gamma_s$ is a step-size and $V(\boldsymbol{y}, \boldsymbol{\theta}, \boldsymbol{\xi}) = -\log p(\boldsymbol{\theta}) - \log p(\boldsymbol{y}|\boldsymbol{\theta}, \boldsymbol{\xi})$ is the $p_{\boldsymbol{\xi}}$ potential, $p_{\boldsymbol{\xi}}(\boldsymbol{y}, \boldsymbol{\theta}) \propto \exp(-V(\boldsymbol{y}, \boldsymbol{\theta}, \boldsymbol{\xi}))$. The dynamics induced lead to samples from $p_{\boldsymbol{\xi}}$ for $s \to \infty$. In the following, we will thus use the notation $\Sigma_s^{\boldsymbol{Y}, \boldsymbol{\theta}}$ to mean that we have access to samples from $p^{(s+1)}$, which is equivalent to apply $\Sigma_s^{\boldsymbol{Y}, \boldsymbol{\theta}}$ to an empirical version of $p^{(s)}$ built from samples $\{(\boldsymbol{y}_i^{(s)}, \boldsymbol{\theta}_i^{(s)})\}_{i=1:N}$. Similarly, we can produce samples $\{\boldsymbol{\theta}_j'^{(s+1)}\}_{j=1:M}$ from the pooled posterior $q_{\boldsymbol{\xi},N}$, using its score expression, via the updating,

$$
\boldsymbol{\theta}'^{(s+1)} = \boldsymbol{\theta}'^{(s)} - \gamma_s' \sum_{i=1}^{N} \nu_i \nabla_{\boldsymbol{\theta}} V(\boldsymbol{y}_i, \boldsymbol{\theta}'^{(s)}, \boldsymbol{\xi}) + \sqrt{2\gamma_s'} \boldsymbol{B}_{\boldsymbol{\theta}',s} .
\tag{17}
$$

For a sampling operator of the pooled posterior general form (13), we need to extend the definition in Marion et al. (2025) by adding a dependence on some distribution $\rho \in \mathcal{P}(\mathcal{Y})$ for the conditioning part. The second sampling operator is defined, for some given $\boldsymbol{\xi}$ and $\rho$, through a sequence over $s$ of parameterized functions from $\mathcal{P}(\boldsymbol{\Theta})$ to $\mathcal{P}(\boldsymbol{\Theta})$ and denoted by $\Sigma_s^{\boldsymbol{\theta}'}(q, \boldsymbol{\xi}, \rho)$. The sampling operator is defined as the outcome of the following process starting from $q^{(0)} \in \mathcal{P}(\boldsymbol{\Theta})$ and iterating

$$
q^{(s+1)} = \Sigma_s^{\boldsymbol{\theta}'}(q^{(s)}, \boldsymbol{\xi}, \rho) .
\tag{18}
$$

For instance, when $\rho = \sum_{i=1}^{N} \nu_i \delta_{\boldsymbol{y}_i}$, $q^{(s+1)} = \Sigma_s^{\boldsymbol{\theta}'}(q^{(s)}, \boldsymbol{\xi}, \rho)$ can then be a shorthand for (17).

## 5 SINGLE LOOP CONTRASTIVE EIG OPTIMIZATION

The perspective of optimization through sampling leads naturally to a nested loop procedure. An inner loop is performed to reach good approximations of $p_{\boldsymbol{\xi}}$ and $q_{\boldsymbol{\xi},N}$ using two samplers as specified in Section 4 and summarized in the nested loop Algorithm 1. Considering sampling as an optimization over the space of distributions (Marion et al., 2025), a more efficient single loop procedure can be derived. As illustrated in the single loop Algorithm 2, at each optimization step over $\boldsymbol{\xi}$, the sampling operators are applied only once using the current $\boldsymbol{\xi}$, which is updated in turn, etc. Sampling operators can be derived from traditional density-based sampling, like in (16) and (17), where an expression of the target distribution is required to compute the score, and also from data-based sampling where only training samples are available. In the latter case, conditional score-based generative models have emerged as a very active field of research. We explicit below how a recent such framework proposed by Dou and Song (2024) can be used in our setting.

**Data-based samplers.** In BOED, we are interested in sampling from a conditional distribution $p(\boldsymbol{\theta}|\boldsymbol{y},\boldsymbol{\xi})$ with the following objectives. First we need to sample from the pooled posterior $q_{\boldsymbol{\xi},N}(\boldsymbol{\theta})$ which requires conditioning on the observation $\boldsymbol{y}$. Second, in sequential design problems (see section 6.1 and Appendix D), we need to condition on the history of observations $\boldsymbol{D}_{k-1}$ and produce samples from $p(\boldsymbol{\theta}|\boldsymbol{D}_{k-1})$. Both these issues can be tackled in the framework of diffusion models for inverse problems (Daras et al., 2024). When the likelihood corresponds to a linear measurement $\boldsymbol{Y}$ with $\boldsymbol{Y} = \boldsymbol{A}_{\boldsymbol{\xi}}\boldsymbol{\theta} + \boldsymbol{\eta}$ and $\boldsymbol{\eta} \sim \mathcal{N}(\mathbf{0},\boldsymbol{\Sigma})$, inspiring recent attempts, such as (Corenflos et al., 2025; Cardoso et al., 2024), have addressed the problem of sampling efficiently from $p(\boldsymbol{\theta}|\boldsymbol{y},\boldsymbol{\xi})$ using only the pre-trained score $s_\phi(\boldsymbol{\theta},t)$ of a diffusion model, without the need for any kind of retraining (see Appendix C for details). For conditional sampling, this would mean running an SDE with a conditional score $\nabla_{\boldsymbol{\theta}} \log p_t(\boldsymbol{\theta}^{(t)}|\boldsymbol{y},\boldsymbol{\xi})$, which is intractable. See (43) and Appendix C.2. As a solution, Dou and Song (2024) propose a method named FPS that approximates $p_t(\boldsymbol{\theta}^{(t)}|\boldsymbol{y},\boldsymbol{\xi})$ by $p_t(\boldsymbol{\theta}^{(t)}|\boldsymbol{y}^{(t)},\boldsymbol{\xi})$ with $\boldsymbol{y}^{(t)}$ the noised observation at time $t$. As the score $\nabla_{\boldsymbol{\theta}}\log p_t(\boldsymbol{\theta}^{(t)}|\boldsymbol{y}^{(t)},\boldsymbol{\xi})$ can be written as $\nabla_{\boldsymbol{\theta}}\log p_t(\boldsymbol{\theta}^{(t)})+\nabla_{\boldsymbol{\theta}}\log p_t(\boldsymbol{y}^{(t)}|\boldsymbol{\theta}^{(t)},\boldsymbol{\xi})$, we can leverage the learned score $s_\phi(\boldsymbol{\theta}^{(t)},t)$ and the closed form of $\nabla_{\boldsymbol{\theta}} \log p_t(\boldsymbol{y}^{(t)}|\boldsymbol{\theta}^{(t)},\boldsymbol{\xi})$ to sample approximately from $p(\boldsymbol{\theta}|\boldsymbol{y},\boldsymbol{\xi})$ using a backward SDE with the approximate score, see (45) in Appendix C.2. This allows to sample efficiently from $p(\boldsymbol{\theta}|\boldsymbol{D}_{k-1})$ and, using $\nabla_{\boldsymbol{\theta}} \log q_{\boldsymbol{\xi},N}(\boldsymbol{\theta}')=\sum_{i=1}^N \nu_i \nabla_{\boldsymbol{\theta}} \log p(\boldsymbol{\theta}'|\boldsymbol{y}_i,\boldsymbol{\xi})$, from the pooled posterior with the extension of (45) below, where $\boldsymbol{y}_i^{(t)}$ is the noised $\boldsymbol{y}_i$ at time $t$ of the forward SDE,

$$d\boldsymbol{\theta}'^{(t)} = \left[ -\frac{\beta(t)}{2}\boldsymbol{\theta}'^{(t)} - \beta(t)\sum_{i=1}^N \nu_i \nabla_{\boldsymbol{\theta}} \log p_t(\boldsymbol{\theta}'^{(t)}|\boldsymbol{y}_i^{(t)},\boldsymbol{\xi}) \right] dt + \sqrt{\beta(t)}d\boldsymbol{B}_t , \quad (19)$$

where $t$ above is now flowing backwards from infinity to $t$=0. In practice, (19) is solved approximately using a numerical discretization and an initialization of the process with $\boldsymbol{\theta}'^{(T)} \sim \mathcal{N}(\mathbf{0},\boldsymbol{I})$ for some large finite $T$. An additional resampling SMC-like step can also be added as explained in Appendix E. The approach allows to handle new sequential data-based BOED tasks as illustrated in Section 6.3.

---

**Algorithm 1:** Nested-loop optimization

**Result:** Optimal design $\boldsymbol{\xi}^*$
**Initialisation:** $\boldsymbol{\xi}_0 \in \mathbb{R}^d$
**for** *t=0:T-1 (outer $\boldsymbol{\xi}$ optimization loop)* **do**
  $p_t^{(0)} \leftarrow p_0$ and $q_t^{(0)} \leftarrow q_0$
  **for** *s=0:S-1 ($p_{\boldsymbol{\xi}}$ inner sampling)* **do**
    $p_t^{(s+1)} = \Sigma_s^{\boldsymbol{Y},\boldsymbol{\theta}}(p_t^{(s)},\boldsymbol{\xi}_t)$
  **end**
  $\hat{p}_{\boldsymbol{\xi}_t} \leftarrow p_t^{(S)}$
  $\hat{\rho}_t \leftarrow \hat{p}_{\boldsymbol{\xi}_t}(\boldsymbol{y})$ ($\hat{p}_{\boldsymbol{\xi}_t}$ marginal over $\boldsymbol{y}$)
  **for** *s'=1:S'-1 ($q_{\boldsymbol{\xi},\rho}$ inner sampling)* **do**
    $q_t^{(s'+1)} = \Sigma_{s'}^{\boldsymbol{\theta}'}(q_t^{(s')},\boldsymbol{\xi}_t,\hat{\rho}_t)$
  **end**
  $\hat{q}_{\boldsymbol{\xi}_t} \leftarrow q_t^{(S')}$
  Compute $\nabla_{\boldsymbol{\xi}} I(\boldsymbol{\xi}_t)=\Gamma(\hat{p}_{\boldsymbol{\xi}_t},\hat{q}_{\boldsymbol{\xi}_t},\boldsymbol{\xi}_t)$ in (12)
  Update $\boldsymbol{\xi}_t$ with SGD or another optimizer
**end**
**return** $\boldsymbol{\xi}_T$;

**Algorithm 2:** Single loop optimization

**Result:** Optimal design $\boldsymbol{\xi}^*$
**Initialisation:** $\boldsymbol{\xi}_0 \in \mathbb{R}^d$, $p^{(0)} \leftarrow p_0$, $q^{(0)} \leftarrow q_0$
**for** *t=0:T-1 (sampling-optimization loop)* **do**
  $p^{(t+1)} = \Sigma_t^{\boldsymbol{Y},\boldsymbol{\theta}}(p^{(t)},\boldsymbol{\xi}_t)$
  $\hat{\rho}_{t+1} \leftarrow p_{\boldsymbol{y}}^{(t+1)}$ ($p^{(t+1)}$ marginal over $\boldsymbol{y}$)
  $q^{(t+1)} = \Sigma_t^{\boldsymbol{\theta}'}(q^{(t)},\boldsymbol{\xi}_t,\hat{\rho}_{t+1})$
  Compute
    $\nabla_{\boldsymbol{\xi}} I(\boldsymbol{\xi}_t)=\Gamma(p^{(t+1)},q^{(t+1)},\boldsymbol{\xi}_t)$ in (12)
  Update $\boldsymbol{\xi}_t$ with SGD or another optimizer
**end**
**return** $\boldsymbol{\xi}_T$;

| Measure | 1 | 2 | 3 | 4 | 5 | 6 |
|---|---|---|---|---|---|---|
| CoDiff | .227 | .338 | .528 | .673 | .789 | .826 |
| Random | .168 | .275 | .350 | .391 | .421 | .463 |

Table 1: CoDiff and random reconstruction quality comparison with SSIM, in [-1,1], the higher the better.

---

**Density-based samplers.** Among density-based samplers, we can mention score-based MCMC samplers, including Langevin dynamics via the Unadjusted Langevin Algorithm (ULA) and Metropolis Adjusted Langevin Algorithm (MALA) (Roberts and Tweedie, 1996), Hamiltonian Monte Carlo (HMC) samplers (Hoffman and Gelman, 2014). In Section 6.2, an illustration is given with Langevin and sequential Monte Carlo (SMC) to handle a sequential density-based BOED task.

**Contrastive Optimization.** Optimizing $\boldsymbol{\xi}$ using the gradient expression (10) encourages to select a $\boldsymbol{\xi}$ that gives either high probability $p(\boldsymbol{y}_i|\boldsymbol{\theta}_i,\boldsymbol{\xi})$ to samples $(\boldsymbol{\theta}_i,\boldsymbol{y}_i)$ from $p_{\boldsymbol{\xi}}$ or low probability

$p(\boldsymbol{y}_j|\boldsymbol{\theta}'_j, \boldsymbol{\xi})$ to samples $\boldsymbol{\theta}'_j$ from $q_{\boldsymbol{\xi},N}$. This contrastive behaviour is also visible in (2) where the EIG is defined as the mean over the experiment outcomes of the KL between posterior and prior distributions. The pooled posterior $q_{\boldsymbol{\xi},N}$ is then used as a proxy to the intractable posterior, to perform this contrastive optimization. Figure 2 provides a visualization of this contrastive behavior in the source localization example of Section 6.2. It corresponds to set the next design $\boldsymbol{\xi}$ to a value that eliminates the most parameter $\boldsymbol{\theta}$ values (right plot) among the possible ones a priori (left plot). This is analogous to Noise Constrastive Estimation (Gutmann and Hyvärinen, 2010) methods where model parameters are computed so that the data samples are as different as possible from the noise samples. Additional illustrations are given in Appendix Figure 6.

# 6 NUMERICAL EXPERIMENTS

Two sequential density-based (Section 6.2) and data-based (Section 6.3) BOED examples are considered to illustrate that our method extends to the sequential case in both settings.

## 6.1 SEQUENTIAL BAYESIAN EXPERIMENTAL DESIGN

In the sequential setting, a sequence of $K$ experiments is planned while gradually accounting for the successively collected data. At step $k$, we wish to pick the best design $\boldsymbol{\xi}_k$ given previous outcomes $\boldsymbol{D}_{k-1} = \{(\boldsymbol{y}_1, \boldsymbol{\xi}_1), \ldots, (\boldsymbol{y}_{k-1}, \boldsymbol{\xi}_{k-1})\}$. The expected information gain in this scenario is given by:

$$I_k(\boldsymbol{\xi}, \boldsymbol{D}_{k-1}) = \mathbb{E}_{p(\boldsymbol{y}|\boldsymbol{\xi}, \boldsymbol{D}_{k-1})}\left[\text{KL}(p(\boldsymbol{\theta}|\boldsymbol{Y}, \boldsymbol{\xi}, \boldsymbol{D}_{k-1}), p(\boldsymbol{\theta}|\boldsymbol{D}_{k-1}))\right] ,$$

where $p(\boldsymbol{\theta}|\boldsymbol{D}_{k-1})$ and $p(\boldsymbol{\theta}|\boldsymbol{y}, \boldsymbol{\xi}, \boldsymbol{D}_{k-1})$ act respectively as prior and posterior analogues to the static case (2). See Appendix D for more detailed explanations. The main difference is that we no longer have direct access to samples from the step $k$ prior $p(\boldsymbol{\theta}|\boldsymbol{D}_{k-1})$. However, as $p(\boldsymbol{\theta}|\boldsymbol{D}_{k-1}) \propto p(\boldsymbol{\theta})\prod_{n=1}^{k-1} p(\boldsymbol{y}_n|\boldsymbol{\theta}, \boldsymbol{\xi}_n)$ and $p(\boldsymbol{\theta}|\boldsymbol{y}, \boldsymbol{\xi}, \boldsymbol{D}_{k-1}) \propto p(\boldsymbol{\theta})p(\boldsymbol{y}|\boldsymbol{\theta}, \boldsymbol{\xi})\prod_{n=1}^{k-1} p(\boldsymbol{y}_n|\boldsymbol{\theta}, \boldsymbol{\xi}_n)$ we can still compute the score of these distributions and run sampling operators similar to (15) and (18). To emphasize their dependence on $\boldsymbol{D}_{k-1}$, they are denoted by $\Sigma_s^{\boldsymbol{Y}, \boldsymbol{\theta}|\boldsymbol{D}_{k-1}}(p^{(s)}, \boldsymbol{\xi})$ and $\Sigma_s^{\boldsymbol{\theta}'|\boldsymbol{D}_{k-1}}(q^{(s)}, \boldsymbol{\xi}, \rho)$. Examples of these operators are provided in (20) and (21) in Section 6.2.

**Evaluation metrics and comparison.** We refer to our method as CoDiff. In Section 6.2, comparison is provided with other recent approaches, namely a reinforcement learning-based approach RL-BOED from Blau et al. (2022), the *variational prior contrastive estimation* VPCE of Foster et al. (2020) and a recent approach named PASOA (Iollo et al., 2024) based on tempered sequential Monte Carlo samplers. We also compare with a non tempered version of this latter approach (SMC) and with a random baseline, where the observations $\{\boldsymbol{y}_1, \cdot, \boldsymbol{y}_K\}$ are simulated with designs generated randomly. More details about these methods are given in Appendix F.2. To compare methods in terms of information gains, we use the *sequential prior contrastive estimation* (SPCE) and *sequential nested Monte Carlo* (SNMC) bounds introduced in Foster et al. (2021) and used in Blau et al. (2022). These quantities allow to compare methods on the produced design sequences only, via their [SPCE, SNMC] intervals which contain the total EIG. Their expressions are given in Appendix F.1. We also provide the $L_2$ Wasserstein distance between the produced samples and the true parameter $\boldsymbol{\theta}$. For methods that do not provide posterior estimations or poor quality ones (RL-BOED, VPCE, Random), we compute Wasserstein distances on posterior samples obtained by using tempered SMC on their design and observation sequences. In contrast, SMC Wasserstein distances are computed on the SMC posterior samples. In Section 6.3, our evaluation is mainly qualitative. The previous methods do not apply and we are not aware of existing attempts that could handle such a generative setting.

## 6.2 SOURCES LOCATION FINDING

We present a source localization example inspired by Foster et al. (2021); Blau et al. (2022). The setup involves $C$ sources in $\mathbb{R}^2$, with unknown positions $\boldsymbol{\theta} = \{\boldsymbol{\theta}_1, \ldots, \boldsymbol{\theta}_C\}$. The challenge is to determine optimal measurement locations to accurately infer the sources positions. When a measurement is taken at location $\boldsymbol{\xi} \in \mathbb{R}^2$, the signal strength is defined as $\mu(\boldsymbol{\theta}, \boldsymbol{\xi}) = b + \sum_{c=1}^{C} \frac{\alpha_c}{m + \|\boldsymbol{\theta}_c - \boldsymbol{\xi}\|_2^2}$ where $\alpha_c$, $b$, and $m$ are predefined constants. We assume a standard Gaussian prior for each source location, $\boldsymbol{\theta}_c \sim \mathcal{N}(0, \boldsymbol{I}_2)$, and model the likelihood as log-normal: $(\log \boldsymbol{y} \mid \boldsymbol{\theta}, \boldsymbol{\xi}) \sim \mathcal{N}(\log \mu(\boldsymbol{\theta}, \boldsymbol{\xi}), \sigma)$, with $\sigma$ representing the standard deviation. For this experiment, we set $C = 2$, $\alpha_1 = \alpha_2 = 1$, $m = 10^{-4}$,

$b = 10^{-1}$, $\sigma = 0.5$, and plan $K = 30$ sequential design optimizations. In the notation of the single loop Algorithm 2, we consider $\Sigma_t^{\boldsymbol{Y}, \boldsymbol{\theta}|\boldsymbol{D}_{k-1}}(p^{(t)}, \boldsymbol{\xi}_t)$ and $\Sigma_t^{\boldsymbol{\theta}'|\boldsymbol{D}_{k-1}}(q^{(t)}, \boldsymbol{\xi}_t, \hat{\rho}_{t+1})$ operators that correspond respectively to the update of batch samples of size $N = 200$ and $M = 200$ $\{(\boldsymbol{y}_i^{(t)}, \boldsymbol{\theta}_i^{(t)})\}_{i=1:N}$ and $\{\boldsymbol{\theta}_j'^{(t)}\}_{j=1:M}$ with $\hat{\rho}_{t+1} = \sum_{i=1}^N \nu_i \delta_{\boldsymbol{y}_i^{(t+1)}}$ using Langevin diffusions. Making use of the availability of the likelihood in this example, sampling from it, is straightforward and sampling operator iterations simplify into, for $i = 1 : N$ and $j = 1 : M$

$$\boldsymbol{\theta}_i^{(t+1)} = \boldsymbol{\theta}_i^{(t)} + \gamma_t \nabla_{\boldsymbol{\theta}} \log p(\boldsymbol{\theta}_i^{(t)}|\boldsymbol{D}_{k-1}) + \sqrt{2\gamma_t} \boldsymbol{B}_{\boldsymbol{\theta}, t} \quad \text{and} \quad \boldsymbol{y}_i^{(t+1)} \sim p(\boldsymbol{y}|\boldsymbol{\theta}_i^{(t+1)}, \boldsymbol{\xi}_t) \quad (20)$$

$$\boldsymbol{\theta}_j'^{(t+1)} = \boldsymbol{\theta}_j'^{(t)} + \gamma_t' \sum_{i=1}^N \nu_i \nabla_{\boldsymbol{\theta}} \log p(\boldsymbol{\theta}_j'^{(t)}|\boldsymbol{y}_i^{(t+1)}, \boldsymbol{\xi}_t, \boldsymbol{D}_{k-1}) + \sqrt{2\gamma_t'} \boldsymbol{B}_{\boldsymbol{\theta}', t} . \quad (21)$$

In practice, Langevin diffusion can get trapped in local minima and causes the sampling to be too slow to keep pace with the optimization process. To address this, we augment the Langevin diffusion with the Diffusive Gibbs (DiGS) MCMC kernel proposed by Chen et al. (2024). DiGS is an auxiliary variable MCMC method where the auxiliary variable $\tilde{\boldsymbol{X}}$ is a noisy version of the original variable $\boldsymbol{X}$. DiGS enhances mixing and helps escape local modes by alternately sampling from the distributions $p(\tilde{x}|x)$, which introduces noise via Gaussian convolution, and $p(x|\tilde{x})$, which denoises the sample back to the original space using a score-based update (here a Langevin diffusion). With 400 total samples, each measurement step takes 2.9 s. This number of samples is insightful as it is usually the amount of samples one can afford to compute in the diffusion models of Section 6.3. The whole experiment is repeated 100 times with random source locations each time. Figure 3 shows, with respect to $k$, the median for SPCE, the $L_2$ Wasserstein distances between weighted samples and the true source locations and SNMC. CoDiff clearly outperforms all other methods, with significant improvement, both in terms of information gain and posterior estimation. It improves by $30\%$ the non-myopic RL-BOED results on SPCE and provides much higher SNMC. The $L_2$ Wasserstein distance is two order of magnitude lower, suggesting the higher quality of our measurements.

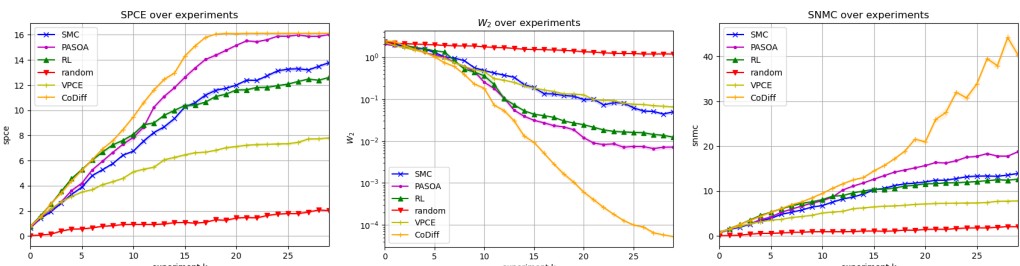

Figure 3: Source location. Median and standard error over 100 rollouts for SPCE, $L_2$ Wasserstein distance (log-scale), SNMC with respect to number of experiments $k$. Number of samples N+M=400.

## 6.3 IMAGE RECONSTRUCTION WITH DIFFUSION MODELS

We build an artificial experimental design task to illustrate the ability of our method to handle design parameters related to inverse problems with a high dimensional parameter $\boldsymbol{\theta}$. We consider the task of recovering an hidden image from only partial observations of its pixels. The image to be recovered is denoted by $\boldsymbol{\theta}$. An experiment corresponds to the choice of a pixel $\boldsymbol{\xi}$ around which an observation mask is centered and the image becomes visible. The measured observation $\boldsymbol{y}$ is then a masked version of $\boldsymbol{\theta}$. The likelihood derives from the model $\boldsymbol{Y} = \boldsymbol{A}_{\boldsymbol{\xi}} \boldsymbol{\theta} + \boldsymbol{\eta}$ where $\boldsymbol{A}_{\boldsymbol{\xi}}$ is a square mask centered at $\boldsymbol{\xi}$ and $\boldsymbol{\eta}$ some Gaussian variable. For the image prior, we consider a diffusion model trained for generation of the MNIST dataset (LeCun et al., 1998). The goal is thus to select sequentially the best central pixel locations for $7 \times 7$ masks so as to reconstruct an entire $28 \times 28$ MNIST image in the smallest number of experiments. The smaller the mask the more interesting it becomes to optimally select the mask centers. Algorithm 2 is used with diffusion-based sampling operators specified in Appendix F.2.2. The gain in optimizing the mask placements is illustrated in Figure 1 and Appendix Figure 9. It is confirmed quantitatively in Table 1, which reports reconstruction quality as measured by the structural similarity index measure (SSIM) (Wang et al., 2004), details in Appendix F.2.2. Progressive reconstructions are shown in Figures 4, 7 and 8. The digit to be recovered is shown in the

1st column. The successively selected masks are shown (red line squares) in the 2nd column with the resulting gradually discovered part of the image. The reconstruction per se can be estimated from the posterior samples shown in the last 16 columns. At each experiment, the upper sub-row shows the 16 most-likely reconstructed images, while the lower sub-row shows the 16 less probable ones. As the number of experiments increases the posterior samples gradually concentrate on the right digit.

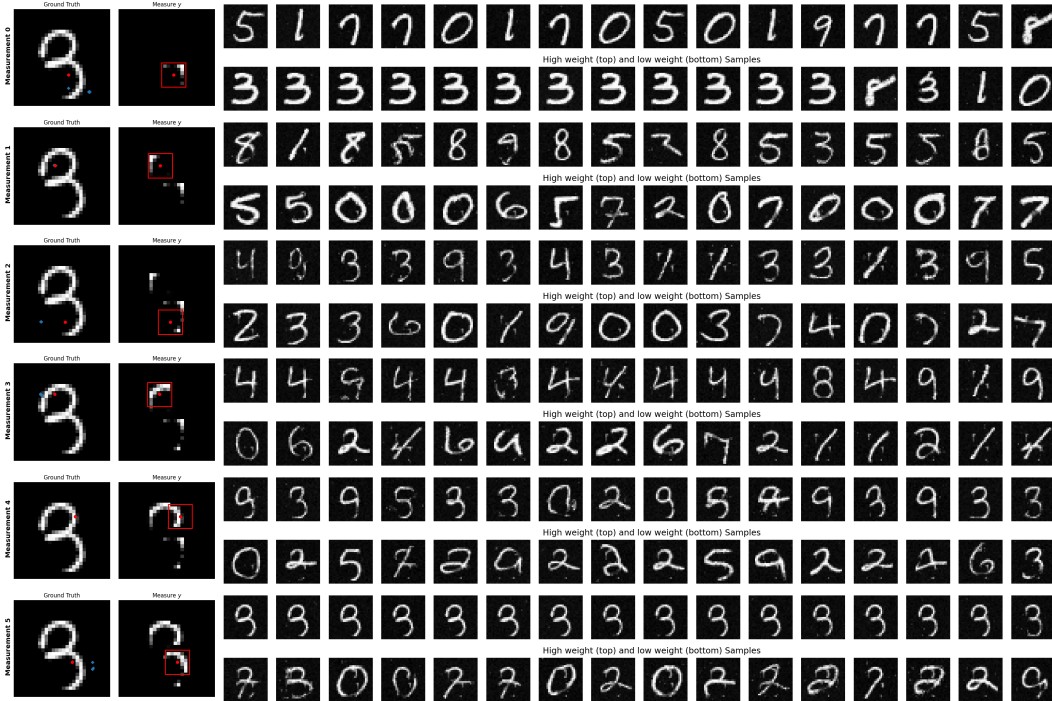

Figure 4: Image reconstruction. First 6 experiments (rows): image ground truth, measurement at experiment $k$, samples from current prior $p(\boldsymbol{\theta}|\boldsymbol{D}_{k-1})$, with best (resp. worst) weights in upper (resp. lower) sub-row. The samples incorporate past measurement information as the procedure advances. Each design steps takes $\sim 7.3$s

## 7 CONCLUSION

We presented a new approach, CoDiff, to gradient-based BOED that allows very efficient implementations. The performance was illustrated in a traditional density-based setting with superior accuracy and lower computational cost compared to state-of-the-art methods. In addition, the possibility of our method to also handle data-based sampling represents, to our knowledge, the first extension of BOED to diffusion-based generative models. By integrating the highly successful framework of diffusion models for our sampling operators, we were able to optimize a design parameter $\boldsymbol{\xi}$ concurrently with the diffusion process. This was illustrated in a new application for BOED involving high dimensional image parameters. The foundation of our approach lies on a new EIG gradient estimator, bi-level optimization, conditional diffusion models and their application to inverse problems. Thanks to this advancement, there are as many new potential applications of BOED as there are trained diffusion models for specific inverse problem tasks. Current limitations include that CoDiff remains a greedy approach, that it requires an explicit expression of the likelihood and that when using diffusions to address inverse problems only linear forward models are currently handled. However, the non-linear setting is an active field of research, and advancements in this area could be directly applied to our framework. The applicability of our method could also be extended by considering settings with no explicit expression of the likelihood and investigating simulation-based inference such as developed by Ivanova et al. (2021); Kleinegesse and Gutmann (2021); Kleinegesse et al. (2020). In addition, although in density-based BOED, we have shown that greedy approaches could outperform long-sighted reinforcement learning procedures, in a data-based setting, it would be interesting to investigate an extension to non myopic approaches such as Iqbal et al. (2024).

ACKNOWLEDGMENTS

The authors thank the reviewers and area chair for their interesting and useful comments and the Inria Challenge project ROAD-AI for partial funding. This work was performed using HPC/AI resources from GENCI-IDRIS (Grant 2023-AD011014217R1). Pierre Alliez is supported by the French government, through the 3IA Côte d'Azur Investments in the Future project managed by the National Research Agency ANR-19-P3IA-0002.

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

## A  TWO EXPRESSIONS FOR THE EIG GRADIENT

Both approaches presented below, that of Goda et al. (2022) and Ao and Li (2024), start from EIG gradient expressions derived using a reparameterization trick. Using the change of variable $\boldsymbol{Y} = T_{\boldsymbol{\xi},\boldsymbol{\theta}}(\boldsymbol{U})$, we can derive the following expression for the EIG gradient,

$$\nabla_{\boldsymbol{\xi}} I(\boldsymbol{\xi}) = \mathbb{E}_{p_U(\boldsymbol{u})p(\boldsymbol{\theta})}\left[\nabla_{\boldsymbol{\xi}} \log p(T_{\boldsymbol{\xi},\boldsymbol{\theta}}(\boldsymbol{U})|\boldsymbol{\theta};\boldsymbol{\xi})\right] - \mathbb{E}_{p_U(\boldsymbol{u})p(\boldsymbol{\theta})}\left[\nabla_{\boldsymbol{\xi}} \log p(T_{\boldsymbol{\xi},\boldsymbol{\theta}}(\boldsymbol{U})|\boldsymbol{\xi})\right] . \tag{22}$$

The first term in (22) involves only the known likelihood and is generally not problematic. For the second term, we can use,

$$\nabla_{\boldsymbol{\xi}} \log p(T_{\boldsymbol{\xi},\boldsymbol{\theta}}(\boldsymbol{U})|\boldsymbol{\xi}) = \frac{\nabla_{\boldsymbol{\xi}} p(T_{\boldsymbol{\xi},\boldsymbol{\theta}}(\boldsymbol{U})|\boldsymbol{\xi})}{p(T_{\boldsymbol{\xi},\boldsymbol{\theta}}(\boldsymbol{U})|\boldsymbol{\xi})} \tag{23}$$

with $p(T_{\boldsymbol{\xi},\boldsymbol{\theta}}(\boldsymbol{U})|\boldsymbol{\xi}) = \mathbb{E}_{p(\boldsymbol{\theta}')}\left[p(T_{\boldsymbol{\xi},\boldsymbol{\theta}}(\boldsymbol{U})|\boldsymbol{\theta}',\boldsymbol{\xi})\right]$ and

$$\begin{aligned}
\nabla_{\boldsymbol{\xi}} p(T_{\boldsymbol{\xi},\boldsymbol{\theta}}(\boldsymbol{U})|\boldsymbol{\xi}) &= \nabla_{\boldsymbol{\xi}} \mathbb{E}_{p(\boldsymbol{\theta}')}\left[p(T_{\boldsymbol{\xi},\boldsymbol{\theta}}(\boldsymbol{U})|\boldsymbol{\theta}',\boldsymbol{\xi})\right] \\
&= \mathbb{E}_{p(\boldsymbol{\theta}')}\left[\nabla_{\boldsymbol{\xi}} \, p(T_{\boldsymbol{\xi},\boldsymbol{\theta}}(\boldsymbol{U})|\boldsymbol{\theta}',\boldsymbol{\xi})\right] \tag{24} \\
&= \mathbb{E}_{p(\boldsymbol{\theta}')}\left[p(T_{\boldsymbol{\xi},\boldsymbol{\theta}}(\boldsymbol{U})|\boldsymbol{\theta}',\boldsymbol{\xi}) \, \nabla_{\boldsymbol{\xi}} \log p(T_{\boldsymbol{\xi},\boldsymbol{\theta}}(\boldsymbol{U})|\boldsymbol{\theta}',\boldsymbol{\xi})\right] . \tag{25}
\end{aligned}$$

Subsequently, two expressions of the EIG gradient can be derived depending on which of (24) or (25) is used. Using (24) and $p(T_{\boldsymbol{\xi},\boldsymbol{\theta}}(\boldsymbol{U})|\boldsymbol{\xi}) = \mathbb{E}_{p(\boldsymbol{\theta}')}\left[p(T_{\boldsymbol{\xi},\boldsymbol{\theta}}(\boldsymbol{U})|\boldsymbol{\theta}',\boldsymbol{\xi})\right]$, it comes

$$\begin{aligned}
\nabla_{\boldsymbol{\xi}} I(\boldsymbol{\xi}) &= \mathbb{E}_{p_U(\boldsymbol{u})p(\boldsymbol{\theta})}\left[\nabla_{\boldsymbol{\xi}} \log p(T_{\boldsymbol{\xi},\boldsymbol{\theta}}(\boldsymbol{U})|\boldsymbol{\theta};\boldsymbol{\xi}) - \frac{\mathbb{E}_{p(\boldsymbol{\theta}')}\left[\nabla_{\boldsymbol{\xi}} \, p(T_{\boldsymbol{\xi},\boldsymbol{\theta}}(\boldsymbol{U})|\boldsymbol{\theta}',\boldsymbol{\xi})\right]}{\mathbb{E}_{p(\boldsymbol{\theta}')}\left[p(T_{\boldsymbol{\xi},\boldsymbol{\theta}}(\boldsymbol{U})|\boldsymbol{\theta}',\boldsymbol{\xi})\right]}\right] \\
&= \mathbb{E}_{p_{\boldsymbol{\xi}}}\left[g(\boldsymbol{\xi},\boldsymbol{Y},\boldsymbol{\theta},\boldsymbol{\theta}) - \frac{\mathbb{E}_{p(\boldsymbol{\theta}')}\left[h(\boldsymbol{\xi},\boldsymbol{Y},\boldsymbol{\theta},\boldsymbol{\theta}')\right]}{\mathbb{E}_{p(\boldsymbol{\theta}')}\left[p(\boldsymbol{Y}|\boldsymbol{\theta}',\boldsymbol{\xi})\right]}\right] . \tag{26}
\end{aligned}$$

with

$$g(\boldsymbol{\xi},\boldsymbol{y},\boldsymbol{\theta},\boldsymbol{\theta}') = \nabla_{\boldsymbol{\xi}} \log p(T_{\boldsymbol{\xi},\boldsymbol{\theta}}(\boldsymbol{u})|\boldsymbol{\theta}',\boldsymbol{\xi})_{|\boldsymbol{u}=T_{\boldsymbol{\xi},\boldsymbol{\theta}}^{-1}(\boldsymbol{y})}$$

and

$$h(\boldsymbol{\xi},\boldsymbol{y},\boldsymbol{\theta},\boldsymbol{\theta}') = \nabla_{\boldsymbol{\xi}} p(T_{\boldsymbol{\xi},\boldsymbol{\theta}}(\boldsymbol{u})|\boldsymbol{\theta}',\boldsymbol{\xi})_{|\boldsymbol{u}=T_{\boldsymbol{\xi},\boldsymbol{\theta}}^{-1}(\boldsymbol{y})}.$$

Considering, in the second term, an additional importance distribution $q(\boldsymbol{\theta}'|\boldsymbol{y},\boldsymbol{\theta},\boldsymbol{\xi})$ leads to the expression used in Goda et al. (2022),

$$\nabla_{\boldsymbol{\xi}} I(\boldsymbol{\xi}) = \mathbb{E}_{p_{\boldsymbol{\xi}}}\left[g(\boldsymbol{\xi},\boldsymbol{Y},\boldsymbol{\theta},\boldsymbol{\theta}) - \frac{\mathbb{E}_{q(\boldsymbol{\theta}'|\boldsymbol{Y},\boldsymbol{\theta},\boldsymbol{\xi})}\left[\frac{p(\boldsymbol{\theta}')}{q(\boldsymbol{\theta}'|\boldsymbol{Y},\boldsymbol{\theta},\boldsymbol{\xi})} h(\boldsymbol{\xi},\boldsymbol{Y},\boldsymbol{\theta},\boldsymbol{\theta}')\right]}{\mathbb{E}_{q(\boldsymbol{\theta}'|\boldsymbol{Y},\boldsymbol{\theta},\boldsymbol{\xi})}\left[\frac{p(\boldsymbol{\theta}')}{q(\boldsymbol{\theta}'|\boldsymbol{Y},\boldsymbol{\theta},\boldsymbol{\xi})} p(\boldsymbol{Y}|\boldsymbol{\theta}',\boldsymbol{\xi})\right]}\right] . \tag{27}$$

It can be used to derive estimators of the form,

$$\nabla_{\boldsymbol{\xi}} I(\boldsymbol{\xi}) \approx \frac{1}{N}\sum_{i=1}^{N}\left[g(\boldsymbol{\xi},\boldsymbol{y}_i,\boldsymbol{\theta}_i,\boldsymbol{\theta}_i) - \frac{\frac{1}{M}\sum_{j=1}^{M}\frac{p(\boldsymbol{\theta}'_{i,j})}{q(\boldsymbol{\theta}'_{i,j}|\boldsymbol{y}_i,\boldsymbol{\theta}_i,\boldsymbol{\xi})} h(\boldsymbol{\xi},\boldsymbol{y}_i,\boldsymbol{\theta}_i,\boldsymbol{\theta}'_{i,j})}{\frac{1}{M}\sum_{j=1}^{M}\frac{p(\boldsymbol{\theta}'_{i,j})}{q(\boldsymbol{\theta}'_{i,j}|\boldsymbol{y}_i,\boldsymbol{\theta}_i,\boldsymbol{\xi})} p(\boldsymbol{y}_i|\boldsymbol{\theta}'_{i,j},\boldsymbol{\xi})}\right] , \tag{28}$$

where $\{(\boldsymbol{y}_i,\boldsymbol{\theta}_i)\}_{i=1:N}$ are simulated from the joint distribution $p_{\boldsymbol{\xi}}$ and for each $i = 1 : N$, $\{\boldsymbol{\theta}'_{i,j}\}_{j=1:M}$ is a sample from $q(\cdot|\boldsymbol{y}_i,\boldsymbol{\theta}_i,\boldsymbol{\xi})$. Goda et al. (2022) use (28) with $N = 1$. Even with perfect sampling, this estimator is not unbiased due to the ratio in the second term but can be de-biased following Rhee and Glynn (2015). The randomized MLMC procedure of Rhee and Glynn (2015) is a post-hoc general procedure that can be more generally applied to de-bias a sequence of possibly biased estimators, provided the estimators are consistent.

Alternatively, using (25) instead, another expression of the EIG gradient can be derived. Replacing (25) in (23), it comes,

$$\begin{aligned}
\nabla_{\boldsymbol{\xi}} \log p(T_{\boldsymbol{\xi},\boldsymbol{\theta}}(\boldsymbol{U})|\boldsymbol{\xi}) &= \mathbb{E}_{p(\boldsymbol{\theta}')}\left[\frac{p(T_{\boldsymbol{\xi},\boldsymbol{\theta}}(\boldsymbol{U})|\boldsymbol{\theta}',\boldsymbol{\xi}) \, \nabla_{\boldsymbol{\xi}} \log p(T_{\boldsymbol{\xi},\boldsymbol{\theta}}(\boldsymbol{U})|\boldsymbol{\theta}',\boldsymbol{\xi})}{p(T_{\boldsymbol{\xi},\boldsymbol{\theta}}(\boldsymbol{U})|\boldsymbol{\xi})}\right] \\
&= \mathbb{E}_{p(\boldsymbol{\theta}'|T_{\boldsymbol{\xi},\boldsymbol{\theta}}(\boldsymbol{U}),\boldsymbol{\xi})}\left[\nabla_{\boldsymbol{\xi}} \log p(T_{\boldsymbol{\xi},\boldsymbol{\theta}}(\boldsymbol{U})|\boldsymbol{\theta}',\boldsymbol{\xi})\right] ,
\end{aligned}$$

which, with the definition of $g$ above, leads to

$$\nabla_{\boldsymbol{\xi}} I(\boldsymbol{\xi}) = \mathbb{E}_{p_{\boldsymbol{\xi}}} \left[ g(\boldsymbol{\xi}, \boldsymbol{Y}, \boldsymbol{\theta}, \boldsymbol{\theta}) - \mathbb{E}_{p(\boldsymbol{\theta}'|\boldsymbol{Y}, \boldsymbol{\xi})} \left[ g(\boldsymbol{\xi}, \boldsymbol{Y}, \boldsymbol{\theta}, \boldsymbol{\theta}') \right] \right] \ . \tag{29}$$

This alternative expression (29) is the starting point of Ao and Li (2024), who subsequently use the following estimator,

$$\nabla_{\boldsymbol{\xi}} I(\boldsymbol{\xi}) \approx \frac{1}{N} \sum_{i=1}^{N} \left[ g(\boldsymbol{\xi}, \boldsymbol{y}_i, \boldsymbol{\theta}_i, \boldsymbol{\theta}_i) - \mathbb{E}_{q(\boldsymbol{\theta}'|\boldsymbol{y}_i, \boldsymbol{\xi})} \left[ g(\boldsymbol{\xi}, \boldsymbol{y}_i, \boldsymbol{\theta}_i, \boldsymbol{\theta}') \right] \right] \ ,$$

where $\{(\boldsymbol{y}_i, \boldsymbol{\theta}_i)\}_{i=1:N}$ is as before a sample from the joint distribution $p_{\boldsymbol{\xi}}$ and where for each $\boldsymbol{y}_i$, $q(\boldsymbol{\theta}'|\boldsymbol{y}_i, \boldsymbol{\xi})$ is a tractable approximation of the intractable posterior $p(\boldsymbol{\theta}'|\boldsymbol{y}_i, \boldsymbol{\xi})$. More specifically, Ao and Li (2024) propose to approximate each posterior distribution by $q(\boldsymbol{\theta}'|\boldsymbol{y}_i, \boldsymbol{\xi}) = \frac{1}{M} \sum_{j=1}^{M} \delta_{\boldsymbol{\theta}'_{i,j}}$, using a sample $\{\boldsymbol{\theta}'_{i,j}\}_{j=1:M}$ from an MCMC procedure. It follows the nested Monte Carlo estimator below,

$$\nabla_{\boldsymbol{\xi}} I(\boldsymbol{\xi}) \approx \frac{1}{N} \sum_{i=1}^{N} \left[ g(\boldsymbol{\xi}, \boldsymbol{y}_i, \boldsymbol{\theta}_i, \boldsymbol{\theta}_i) - \frac{1}{M} \sum_{j=1}^{M} g(\boldsymbol{\xi}, \boldsymbol{y}_i, \boldsymbol{\theta}_i, \boldsymbol{\theta}_{i,j}) \right] \ . \tag{30}$$

## B  LOGARITHMIC POOLING AS A GOOD IMPORTANCE SAMPLING PROPOSAL

When considering importance sampling with a proposal distribution $q$ and a target distribution $p$, Chatterjee and Diaconis (2018) proved that under certain conditions, the number of simulation draws required for both importance sampling and self normalized importance sampling (SNIS) estimators to have small $L_1$ error with high probability was roughly $\exp(\mathrm{KL}(p, q))$, see Theorem 1.2 in Chatterjee and Diaconis (2018) for SNIS. Similarly, selecting a proposal distribution which minimizes the importance sampling estimator variance is equivalent to finding a distribution with small $\chi^2$-distance to $p$, see *e.g.* Appendix E of Minka (2005). More generally, finding a good proposal $q$ is linked to the problem of minimizing $\alpha$-divergences or $f$-divergence between $p$ and $q$, which are jointly convex in $p$ and $q$, see Minka (2005). In this work, we consider $\mathrm{KL}(q, p)$ as a measure of proximity between $p$ and $q$. This choice is ultimately arbitrary but has the advantage of leading to an interpretable proposal with interesting sampling properties. To justify the pooled posterior $q_{\boldsymbol{\xi}, N}$ in (9) and its use in (10), we then use Lemma 2 below to show that for $\sum_{i=1}^{N} \nu_i = 1$, the distribution $q^*$ that minimizes the weighted sum of the KL against each posterior $p(\boldsymbol{\theta}|\boldsymbol{y}_i, \boldsymbol{\xi})$, *i.e.* $\sum_{i=1}^{N} \nu_i \mathrm{KL}(q, p(\boldsymbol{\theta}|\boldsymbol{y}_i, \boldsymbol{\xi}))$ is

$$q^*(\boldsymbol{\theta}) \quad \propto \quad p(\boldsymbol{\theta}) \prod_{i=1}^{N} p(\boldsymbol{y}_i|\boldsymbol{\theta}, \boldsymbol{\xi})^{\nu_i} \tag{31}$$

$$\propto \quad \prod_{i=1}^{N} p(\boldsymbol{\theta}|\boldsymbol{y}_i, \boldsymbol{\xi})^{\nu_i} \ , \tag{32}$$

which is the logarithmic pooling (or geometric mixture) of the respective posterior distributions $p(\boldsymbol{\theta}|\boldsymbol{y}_i, \boldsymbol{\xi})$. Lemma 2 results from an application of a lemma mentioned by Alquier (2024) (Lemma 2.2 therein), and recalled below in Lemma 1. This Lemma 1 has been known since Kullback (Kullback, 1959) in the case of a finite parameter space $\boldsymbol{\Theta}$, but the general case is due to Donsker and Varadhan (Donsker and Varadhan, 1976). Recall that $\mathcal{P}(\boldsymbol{\Theta})$ denotes the set of probability measures on $\boldsymbol{\Theta}$ and $p$ a given probability measure in $\mathcal{P}(\boldsymbol{\Theta})$.

**Lemma 1 (Donsker and Varadhan's variational formula)** *For any measurable, bounded function $f : \boldsymbol{\Theta} \to \mathbb{R}$, the supremum with respect to $q \in \mathcal{P}(\boldsymbol{\Theta})$ of*

$$\mathbb{E}_q \left[ f(\boldsymbol{\theta}) \right] - KL(q, p)$$

*is the following Gibbs measure $p_f$ defined by its density with respect to $p$,*

$$dp_f = \frac{\exp(f(\boldsymbol{\theta}))}{\mathbb{E}_p \left[ \exp(f(\boldsymbol{\theta})) \right]} dp \ .$$

The following Lemma 2 is an application of Lemma 1.

**Lemma 2** *For a given probability measure $p \in \mathcal{P}(\Theta)$ and a measure $\rho$ on $\mathcal{Y}$ (not necessarily a probability measure), define for any probability measure $q \in \mathcal{P}(\Theta)$*

$$\ell(q) = \mathbb{E}_q\left[\mathbb{E}_{\boldsymbol{Y}\sim\rho}\left[\log p(\boldsymbol{Y}|\boldsymbol{\theta})\right]\right] - KL(q, p) . \tag{33}$$

*It results from the Donsker and Varadhan's variational formula Lemma 1 that the supremum of $\ell(q)$ with respect to $q$ is reached for the Gibbs measure $q^*$ defined by its density with respect to $p$,*

$$q^*(\boldsymbol{\theta}) \propto p(\boldsymbol{\theta}) \, \exp(\mathbb{E}_{Y\sim\rho}[\log p(\boldsymbol{Y}|\boldsymbol{\theta})]) .$$

*In addition maximizing $\ell$ is equivalent to minimising*

$$\mathbb{E}_{\boldsymbol{Y}\sim\rho}\left[KL(q(\boldsymbol{\theta}), p(\boldsymbol{\theta}|\boldsymbol{Y}))\right]$$

*which means that $q^*$ is the measure that minimizes the KL to each $p(\boldsymbol{\theta}|\boldsymbol{y})$ on average with respect to $\boldsymbol{y}$.*

**Proof of Lemma 2** The expression of $q^*$ results from a direct application of Lemma 1 to $f(\boldsymbol{\theta}) = \mathbb{E}_{\boldsymbol{Y}\sim\rho}[\log p(\boldsymbol{Y}|\boldsymbol{\theta})]$ assuming it is measurable and bounded as a function of $\boldsymbol{\theta}$ (to be checked in practice). The second part results from rewriting $\ell$ as

$$\begin{aligned}
\ell(q) &= \mathbb{E}_{\boldsymbol{Y}\sim\rho}\left[\mathbb{E}_{\boldsymbol{\theta}\sim q}\left[\frac{\log(p(\boldsymbol{Y}|\boldsymbol{\theta})p(\boldsymbol{\theta}))}{\log q(\boldsymbol{\theta})}\right]\right] \\
&= -\mathbb{E}_{\boldsymbol{Y}\sim\rho}\left[\mathrm{KL}(q, p(\boldsymbol{\theta}|\boldsymbol{Y})\right] + \mathbb{E}_{\boldsymbol{Y}\sim\rho}\left[\log p(\boldsymbol{Y})\right] .
\end{aligned}$$

$\square$

Example: as already mentioned our pooled posterior $q_{\boldsymbol{\xi},N}$ corresponds to the application of this result to $\rho = \sum_{i=1}^N \nu_i \delta_{\boldsymbol{y}_i}$ with $\sum_{i=1}^N \nu_i = 1$. This latter result with $\sum_{i=1}^N \nu_i = 1$ can also be recovered from a more general result by Amari (2007), which is stated for any $\alpha$-divergence (see Theorem 2 of Amari (2007) with $\alpha = 1$). The continuous weight version is also mentioned by Amari (2007) (Theorem 4), referring to other papers for the proof. We provided here a simple proof for the KL ($\alpha = 1$) case.

Remark 1: If $\rho = \delta_{\boldsymbol{y}}$, or $\rho = \sum_{i=1}^N \delta_{\boldsymbol{y}_i}$, we recover the standard variational formulation of the posterior distribution (see *e.g.* Table 1 in (Knoblauch et al., 2022)). The posterior distribution $p(\boldsymbol{\theta}|\boldsymbol{y}_1, \ldots, \boldsymbol{y}_N)$ differs from the logarithmic pooling (for which the weights $\nu_i$ sum to 1) in the relative weight given to the prior. The result is valid for very general $\ell$ not necessarily expressed as an expectation.

Remark 2: Regarding logarithmic pooling, the result is similar to a result in Carvalho et al. (2022) (Remark 3.1 therein) by showing that, in the case of the sum of the KL, $\sum_{i=1}^N \mathrm{KL}(q, p(\boldsymbol{\theta}|\boldsymbol{y}_i))$, the optimal pooling weights are equal, $\nu_i = \frac{1}{N}$.

Remark 3: The pooled posterior distribution can also be recovered as a *constrained* mean field solution. Indeed, it is easy to show that $q^*$ is also the measure that minimizes the KL between the joint distribution and a product form approximation where one of the factor is fixed to $\rho(\boldsymbol{y})$,

$$q^* = \arg\min_{q\in\mathcal{P}(\Theta)} \mathrm{KL}(q(\boldsymbol{\theta})\rho(\boldsymbol{y}), p(\boldsymbol{\theta}, \boldsymbol{Y})) .$$

## C    DIFFUSION-BASED GENERATIVE MODELS

### C.1    DENOISING DIFFUSION MODELS

Given a distribution $p_0 \in \mathcal{P}(\Theta)$ only available through a set of samples of $\boldsymbol{\theta}$, diffusion models are based on the addition of noise to the available samples in such a manner that allows to learn the reverse process that "denoises" the samples. This learned process can then be exploited to generate new samples by denoising random noise samples until we get back to the original data distribution. As an appropriate noising process, in our experiments we ran the Variance Preserving SDE from Dhariwal and Nichol (2021):

$$d\tilde{\boldsymbol{\theta}}^{(t)} = -\frac{\beta(t)}{2}\tilde{\boldsymbol{\theta}}^{(t)}dt + \sqrt{\beta(t)}d\tilde{\boldsymbol{B}}_t \tag{34}$$

where $\beta(t) > 0$ is a linear noise schedule that controls the amount of noise added at time $t$. Solving SDE (34) leads to

$$(\tilde{\boldsymbol{\theta}}^{(t)}|\tilde{\boldsymbol{\theta}}^{(0)}) \sim \mathcal{N}\left(\tilde{\boldsymbol{\theta}}^{(0)}\exp(-\frac{1}{2}\int_0^t \beta(s)ds),\ (1-\exp(-\int_0^t \beta(s)ds))\boldsymbol{I}\right)\ , \tag{35}$$

which can be written as

$$\tilde{\boldsymbol{\theta}}^{(t)} = \sqrt{\bar{\alpha}_t}\tilde{\boldsymbol{\theta}}^{(0)} + \sqrt{1-\bar{\alpha}_t}\boldsymbol{\epsilon} \quad \text{with} \quad \bar{\alpha}_t = \exp(-\int_0^t \beta(s)ds) \quad \text{and} \tag{36}$$

where $\boldsymbol{\epsilon} \sim \mathcal{N}(\boldsymbol{0}, \boldsymbol{I})$ is a standard Gaussian random variable. Samples from $p_0$ are transformed to samples approximately from a standard Gaussian distribution after some large time $T$.

The reverse denoising process can then be written as the reverse of the diffusion process (34), which as stated by Anderson (1982) is:

$$d\boldsymbol{\theta}^{(t)} = \left[-\frac{\beta(t)}{2}\boldsymbol{\theta}^{(t)} - \beta(t)\nabla_{\boldsymbol{\theta}}\log p_t(\boldsymbol{\theta}^{(t)})\right] dt + \sqrt{\beta(t)}d\boldsymbol{B}_t\ , \tag{37}$$

where $t$ flows backwards from infinity to $t=0$ and $p_t$ is the distribution of $\tilde{\boldsymbol{\theta}}^{(t)}$ from (34). In practice, the process is started at some large finite $T$ assuming that $\boldsymbol{\theta}^{(T)} \sim \mathcal{N}(\boldsymbol{0}, \boldsymbol{I})$. In this Appendix, we rather consider increasing time $t$ from 0 to $T$, using that (37) can be equivalently written as,

$$d\boldsymbol{\theta}^{(t)} = \left[\frac{\beta(T-t)}{2}\boldsymbol{\theta}^{(t)} + \beta(T-t)\nabla_{\boldsymbol{\theta}}\log p_{T-t}(\boldsymbol{\theta}^{(t)})\right] dt + \sqrt{\beta(T-t)}d\boldsymbol{B}_t\ , \tag{38}$$

which is now initialized with $\boldsymbol{\theta}^{(0)} \sim \mathcal{N}(\boldsymbol{0}, \boldsymbol{I})$. Solving this reverse SDE, the distribution of $\boldsymbol{\theta}^{(T)}$ is closed to $p_0$ for large $T$, which allows approximate sampling from $p_0$.

The score function $\nabla_{\boldsymbol{\theta}}\log p_t(\boldsymbol{\theta})$ of the noisy data distribution at time $t$ is intractable and is then estimated by learning a neural network $s_\phi(\boldsymbol{\theta}, t)$ with parameters $\phi$. Score matching (Hyvärinen, 2005) is a method to train $s_\phi$ by minimizing the following loss:

$$\mathbb{E}_{p_t(\boldsymbol{\theta})}\left[||s_\phi(\boldsymbol{\theta}, t) - \nabla_{\boldsymbol{\theta}}\log p_t(\boldsymbol{\theta})||^2\right]\ . \tag{39}$$

As $p_t(\boldsymbol{\theta})$ is still unknown and only samples from $p_t(\boldsymbol{\theta}|\tilde{\boldsymbol{\theta}}^{(0)})$ in (35) are available, Song et al. (2021) rewrite this loss function as:

$$\mathbb{E}_{t\sim U[0,T]}\mathbb{E}_{p_0(\boldsymbol{\theta}^{(0)})}\mathbb{E}_{p_t(\boldsymbol{\theta}|\boldsymbol{\theta}^{(0)})}\left[\lambda(t)||s_\phi(\boldsymbol{\theta}, t) - \nabla_{\boldsymbol{\theta}}\log p_t(\boldsymbol{\theta}|\boldsymbol{\theta}^{(0)})||^2\right] \tag{40}$$

where $\lambda(t) > 0$ is a weighting function that allows to focus more on certain timesteps than others. It is common to take $\lambda(t)$ inversely proportional to the variance of (35) at time $t$.

Once the neural network $s_\phi$ has been trained by minimizing (40), it can be used to generate new samples approximately distributed as the target distribution $p_0$ by running a numerical scheme on the reverse SDE (38). By running for example the Euler-Maruyama scheme on (38), we get the following update step for the reverse process:

$$\boldsymbol{\theta}^{(t+\Delta t)} = \boldsymbol{\theta}^{(t)} + \frac{\beta(T-t)}{2}\boldsymbol{\theta}^{(t)}\Delta t + \beta(T-t)s_\phi(\boldsymbol{\theta}^{(t)}, T-t)\Delta t + \sqrt{\beta(T-t)\Delta t}\boldsymbol{\epsilon}\ . \tag{41}$$

We can then generate samples approximately from $p_0$ by running the reverse process (41) with a small enough $\Delta t$.

## C.2 CONDITIONAL DIFFUSION MODELS

Conditional diffusion models arise when, for some measurement $\boldsymbol{y}$, we want to produce samples from some conditional distribution $p_0(\boldsymbol{\theta}|\boldsymbol{y})$. Sampling from conditional distributions is a problem that arises in inverse problems. When using diffusion models, numerous solutions have been investigated as mentioned in a very recent review (Daras et al., 2024). We specify in this section the approach adopted for our applications. With the application to experimental design in mind, we assume here that

$$\boldsymbol{Y} = \boldsymbol{A}_{\boldsymbol{\xi}}\boldsymbol{\theta} + \boldsymbol{\eta} \tag{42}$$

where $\boldsymbol{\eta} \sim \mathcal{N}(\mathbf{0}, \sigma^2 \boldsymbol{I})$ is the measurement noise, $\boldsymbol{A_\xi}$ is the operator that represents the experiment at $\boldsymbol{\xi}$.

Sampling from the conditional distribution $p(\boldsymbol{\theta}|\boldsymbol{y}, \boldsymbol{\xi})$ can be done by running the reverse diffusion process on the conditional SDE:

$$d\boldsymbol{\theta}^{(t)} = \left[\frac{\beta(T-t)}{2}\boldsymbol{\theta}^{(t)} + \beta(T-t)\nabla_{\boldsymbol{\theta}} \log p_{T-t}(\boldsymbol{\theta}^{(t)}|\boldsymbol{y}, \boldsymbol{\xi})\right]dt + \sqrt{\beta(T-t)}d\boldsymbol{B}_t, \quad (43)$$

with the usual score $\nabla_{\boldsymbol{\theta}} \log p_{T-t}(\boldsymbol{\theta}^{(t)})$ replaced by the conditionnal score $\nabla_{\boldsymbol{\theta}} \log p_{T-t}(\boldsymbol{\theta}^{(t)}|\boldsymbol{y}, \boldsymbol{\xi})$. The main objective of conditional SDE is to generate samples from the conditional distribution $p(\boldsymbol{\theta}|\boldsymbol{y}, \boldsymbol{\xi})$ without retraining a new neural network $s_\phi$ for the new conditional score. Writing in terms of the forward process, the conditional score can be written using:

$$\nabla_{\tilde{\boldsymbol{\theta}}} \log p_t(\tilde{\boldsymbol{\theta}}^{(t)}|\boldsymbol{y}, \boldsymbol{\xi}) = \nabla_{\tilde{\boldsymbol{\theta}}} \log p_t(\boldsymbol{y}|\tilde{\boldsymbol{\theta}}^{(t)}, \boldsymbol{\xi}) + \nabla_{\tilde{\boldsymbol{\theta}}} \log p_t(\tilde{\boldsymbol{\theta}}^{(t)}) \quad (44)$$

and we can leverage a pre-computed neural network $s_\phi$ that was trained to estimate the score $\nabla_{\tilde{\boldsymbol{\theta}}} \log p_t(\tilde{\boldsymbol{\theta}}^{(t)})$ in the unconditional case. If we know how to evaluate the first term $\nabla_{\tilde{\boldsymbol{\theta}}} \log p_t(\boldsymbol{y}|\tilde{\boldsymbol{\theta}}^{(t)}, \boldsymbol{\xi})$, we can then run the reverse process (43) to generate samples from the conditional distribution $p(\boldsymbol{\theta}|\boldsymbol{y}, \boldsymbol{\xi})$. Unfortunately, this term does not have a closed form expression. As a solution, Dou and Song (2024) propose to approximate the intractable $\nabla_{\tilde{\boldsymbol{\theta}}} \log p_t(\boldsymbol{y}|\tilde{\boldsymbol{\theta}}^{(t)}, \boldsymbol{\xi})$ by the tractable $\nabla_{\tilde{\boldsymbol{\theta}}} \log p_t(\boldsymbol{y}^{(t)}|\tilde{\boldsymbol{\theta}}^{(t)}, \boldsymbol{\xi})$ where $\boldsymbol{y}^{(t)}$ is a noisy version of $\boldsymbol{y}$ at time $t$. Then, the following backward SDE can be run to generate samples from the conditional distribution $p(\boldsymbol{\theta}|\boldsymbol{y}, \boldsymbol{\xi})$:

$$d\boldsymbol{\theta}^{(t)} = \left[\frac{\beta(T-t)}{2}\boldsymbol{\theta}^{(t)} + \beta(T-t)\nabla_{\theta} \log p_{T-t}(\boldsymbol{y}^{(T-t)}|\boldsymbol{\theta}^{(t)}, \boldsymbol{\xi}) + \beta(T-t)\nabla_{\theta} \log p_{T-t}(\boldsymbol{\theta}^{(t)})\right]dt$$
$$+ \sqrt{\beta(T-t)}d\boldsymbol{B}_t \quad (45)$$

The sequence of noisy $\boldsymbol{y}^{(t)}$ can be generated with a noising process like (36),

$$\boldsymbol{y}^{(t)} = \sqrt{\bar{\alpha}_t}\boldsymbol{y} + \sqrt{1 - \bar{\alpha}_t}\boldsymbol{A_\xi}\boldsymbol{\epsilon} \quad \text{with} \quad \bar{\alpha}_t = \exp(-\int_0^t \beta(s)ds), \quad (46)$$

which using the forward model (42) can be written as:

$$\boldsymbol{y}^{(t)} = \boldsymbol{A_\xi}\tilde{\boldsymbol{\theta}}^{(t)} + \sqrt{\bar{\alpha}_t}\boldsymbol{\eta}. \quad (47)$$

We can then evaluate $\nabla_{\tilde{\boldsymbol{\theta}}} \log p_t(\boldsymbol{y}^{(t)}|\tilde{\boldsymbol{\theta}}^{(t)}, \boldsymbol{\xi})$ as :

$$\nabla_{\tilde{\boldsymbol{\theta}}} \log p_t(\boldsymbol{y}^{(t)}|\tilde{\boldsymbol{\theta}}^{(t)}, \boldsymbol{\xi}) = \frac{1}{\sigma^2 \bar{\alpha}_t}\boldsymbol{A_\xi}^T(\boldsymbol{y}^{(t)} - \boldsymbol{A_\xi}\tilde{\boldsymbol{\theta}}^{(t)}). \quad (48)$$

## C.3 GRADIENT ESTIMATION

When sampling is performed with a finite time horizon diffusion model, the first iterations of the corresponding sampling operator may provide too noisy samples which would result in gradient estimations with little information. One solution proposed by Marion et al. (2025) is to use a queuing trick which requires to store in memory a queue of samples. The memory burden is high and only acceptable for a limited number of particles. We propose a simpler solution, which consists in using Tweedie's formula (Efron, 2011) to perform a one-shot backward step, replacing a potentially too noisy $\boldsymbol{\theta}^{(t)}$ by the conditional mean $\mathbb{E}[\tilde{\boldsymbol{\theta}}^{(0)}|\tilde{\boldsymbol{\theta}}^{(T-t)} = \boldsymbol{\theta}^{(t)}]$, which can be interpreted as its prediction at time 0. The Tweedie's formula provides this prediction in close-form,

$$\hat{\boldsymbol{\theta}}^{(t)} = \mathbb{E}[\tilde{\boldsymbol{\theta}}^{(0)}|\tilde{\boldsymbol{\theta}}^{(T-t)} = \boldsymbol{\theta}^{(t)}] = \frac{\boldsymbol{\theta}^{(t)} + (1 - \bar{\alpha}_{T-t})\nabla_{\boldsymbol{\theta}} \log p_{T-t}(\boldsymbol{\theta}^{(t)})}{\sqrt{\bar{\alpha}_{T-t}}}. \quad (49)$$

Gradients are then computed using the $\hat{\boldsymbol{\theta}}^{(t)}$'s values, while $\boldsymbol{\theta}^{(t)}$ is updated into $\boldsymbol{\theta}^{(t+1)}$ from the backward SDE, as mentioned above. This is only really impactful for small $t$ as for large $t$, $\hat{\boldsymbol{\theta}}^{(t)}$ and $\boldsymbol{\theta}^{(t)}$ get closer. This extra computation does not add cost as (49) uses a score value that is already computed for (45).

## D    SEQUENTIAL BAYESIAN EXPERIMENTAL DESIGN

In this framework, experimental conditions are determined sequentially, making use of measurements that are gradually made. This sequential view is referred to as sequential or iterated design. In a sequential setting, we assume that we plan a sequence of $K$ experiments. For each experiment, we wish to pick the best $\boldsymbol{\xi}_k$ using the data that has already been observed $\boldsymbol{D}_{k-1} = \{(\boldsymbol{y}_1, \boldsymbol{\xi}_1), \ldots, (\boldsymbol{y}_{k-1}, \boldsymbol{\xi}_{k-1})\}$. Given this design, we conduct an experiment using $\boldsymbol{\xi}_k$ and obtain outcome $\boldsymbol{y}_k$. Both $\boldsymbol{\xi}_k$ and $\boldsymbol{y}_k$ are then added to $\boldsymbol{D}_{k-1}$ for a new set $\boldsymbol{D}_k = \boldsymbol{D}_{k-1} \cup (\boldsymbol{y}_k, \boldsymbol{\xi}_k)$. After each step, our belief about $\boldsymbol{\theta}$ is updated and summarised by the current posterior $p(\boldsymbol{\theta}|\boldsymbol{D}_k)$, which acts as the next prior at step $k+1$. When the observations are assumed conditionally independent, it comes,

$$p(\boldsymbol{\theta}|\boldsymbol{D}_k) \propto p(\boldsymbol{\theta}) \prod_{n=1}^{k} p(\boldsymbol{y}_n|\boldsymbol{\theta}, \boldsymbol{\xi}_n) \tag{50}$$

and

$$p(\boldsymbol{y}, \boldsymbol{\theta}|\boldsymbol{\xi}, \boldsymbol{D}_{k-1}) \propto p(\boldsymbol{\theta}) \, p(\boldsymbol{y}|\boldsymbol{\theta}, \boldsymbol{\xi}) \prod_{n=1}^{k-1} p(\boldsymbol{y}_n|\boldsymbol{\theta}, \boldsymbol{\xi}_n) \, . \tag{51}$$

A greedy design can be seen as choosing each design $\boldsymbol{\xi}_k$ as if it was the last one. This means that $\boldsymbol{\xi}_k$ is chosen as $\boldsymbol{\xi}_k^*$ the value that maximizes

$$\boldsymbol{\xi}_k^* = \arg \max_{\boldsymbol{\xi}} I_k(\boldsymbol{\xi}, \boldsymbol{D}_{k-1})$$

where

$$I_k(\boldsymbol{\xi}, \boldsymbol{D}_{k-1}) = \mathbb{E}_{p_{\boldsymbol{\xi}}^k} \left[ \log \frac{p_{\boldsymbol{\xi}}^k(\boldsymbol{\theta}, \boldsymbol{Y})}{p(\boldsymbol{Y}|\boldsymbol{\xi}, \boldsymbol{D}_{k-1}) \, p(\boldsymbol{\theta}|\boldsymbol{D}_{k-1})} \right] = \mathrm{MI}(p_{\boldsymbol{\xi}}^k) \tag{52}$$

with $p_{\boldsymbol{\xi}}^k$ denoting the joint distribution $p(\boldsymbol{y}, \boldsymbol{\theta}|\boldsymbol{\xi}, \boldsymbol{D}_{k-1}) = p(\boldsymbol{y}|\boldsymbol{\theta}, \boldsymbol{\xi}) \, p(\boldsymbol{\theta}|\boldsymbol{D}_{k-1})$. Distribution $p_{\boldsymbol{\xi}}^k$ involves the current prior $p(\boldsymbol{\theta}|\boldsymbol{D}_{k-1})$, which is not available in closed-form and is not straightforward to sample from. Distribution $p_{\boldsymbol{\xi}}^k$ can be written as a Gibbs distribution by defining the potential $V_k$ as

$$p_{\boldsymbol{\xi}}^k(\boldsymbol{y}, \boldsymbol{\theta}) \propto \exp\left(-V_k(\boldsymbol{y}, \boldsymbol{\theta}, \boldsymbol{\xi})\right)$$

$$\text{with } V_k(\boldsymbol{y}, \boldsymbol{\theta}, \boldsymbol{\xi}) = -\log p(\boldsymbol{\theta}) - \log p(\boldsymbol{y}|\boldsymbol{\theta}, \boldsymbol{\xi}) - \sum_{n=1}^{k-1} \log p(\boldsymbol{y}_n|\boldsymbol{\theta}, \boldsymbol{\xi}_n)$$

$$= V(\boldsymbol{y}, \boldsymbol{\theta}, \boldsymbol{\xi}) + \tilde{V}_k(\boldsymbol{\theta}) \, ,$$

where $V(\boldsymbol{y}, \boldsymbol{\theta}, \boldsymbol{\xi})$ has been already defined in Section 4. Note that the marginal in $\boldsymbol{\theta}$ of $p_{\boldsymbol{\xi}}^k$ is the posterior at step $k-1$ or equivalently the current prior $p(\boldsymbol{\theta}|\boldsymbol{D}_{k-1})$ and the marginal in $\boldsymbol{y}$ is

$$p(\boldsymbol{y}|\boldsymbol{\xi}, \boldsymbol{D}_{k-1}) = \mathbb{E}_{p(\boldsymbol{\theta}|\boldsymbol{D}_{k-1})} \left[ p(\boldsymbol{y}|\boldsymbol{\theta}, \boldsymbol{D}_{k-1}) \right] .$$

Once a new $\boldsymbol{\xi}_k$ is computed and a new observation $\boldsymbol{y}_k$ is performed, the posterior at step $k$ is $p(\boldsymbol{\theta}|\boldsymbol{y}_k, \boldsymbol{\xi}_k, \boldsymbol{D}_{k-1})$ which is the conditional distribution of $p(\boldsymbol{y}_k, \boldsymbol{\theta}|\boldsymbol{\xi}_k, \boldsymbol{D}_{k-1})$.

## E    SEQUENTIAL MONTE CARLO (SMC)-STYLE RESAMPLING

SMC is an essential addition when dealing with sequential BOED. In density-based BOED, it has been already exploited in the sequential context showing a real improvement in the quality of the generated samples (Iollo et al., 2024). SMC is also useful in simpler static cases as it can improve the quality of the generated $\boldsymbol{\theta}$ and contrastive $\boldsymbol{\theta}'$ samples, that in turn improves the accuracy of the gradient estimator (30). A particularly central step in SMC is the resampling step, first recalled below in the density-based case. Using our framework, is it also possible to derive a SMC-style resampling scheme in the data-based case. This is becoming a popular strategy in the context of generative models (Dou and Song, 2024; Cardoso et al., 2024).

**Density-based BOED.** In static density-based BOED, the prior $p(\boldsymbol{\theta})$ and the likelihood $p(\boldsymbol{y}|\boldsymbol{\theta},\boldsymbol{\xi})$ are available in closed-form. In the sequential experiment context, we want to generate $N$ samples $\boldsymbol{\theta}_1,\ldots,\boldsymbol{\theta}_N$ from the current prior $p(\boldsymbol{\theta}|\boldsymbol{D}_{k-1})$ and $M$ samples $\boldsymbol{\theta}'_1,\ldots,\boldsymbol{\theta}'_M$ from the pooled posterior $q_{\boldsymbol{\xi},N}(\boldsymbol{\theta}')$. As both $p(\boldsymbol{\theta}|\boldsymbol{D}_{k-1})$ and $q_{\boldsymbol{\xi},N}(\boldsymbol{\theta}') \propto \prod_{i=1}^{N} p(\boldsymbol{\theta}'|\boldsymbol{y}_i,\boldsymbol{\xi},\boldsymbol{D}_{k-1})^{\nu_i}$ can be evaluated up to a normalizing constant, it is straightforward to extend the sampling operators of Section 4 and add a resampling step to the samples $\boldsymbol{\theta}_1,\ldots,\boldsymbol{\theta}_N$ and $\boldsymbol{\theta}'_1,\ldots,\boldsymbol{\theta}'_M$ with weights $w_i$ and $w'_j$:

$$w_i = \frac{\tilde{w}_i}{\sum_{i=1}^{N} \tilde{w}_i} \quad \text{with} \quad \tilde{w}_i = \tilde{p}(\boldsymbol{\theta}_i)$$

$$w'_j = \frac{\tilde{w'_j}}{\sum_{j=1}^{M} \tilde{w'_j}} \quad \text{with} \quad \tilde{w'_j} = \tilde{q}(\boldsymbol{\theta}'_j)$$

where $\tilde{p}$ and $\tilde{q}$ are the unnormalized versions of $p(\boldsymbol{\theta}|\boldsymbol{D}_{k-1})$ and $q_{\boldsymbol{\xi},N}(\boldsymbol{\theta}')$ respectively.

**Data-based BOED.** In the setting of data-based BOED, we assume access to a conditional diffusion model that allows to generate samples from $p(\boldsymbol{\theta}|\boldsymbol{D}_{k-1})$ and $q_{\boldsymbol{\xi},N}(\boldsymbol{\theta}')$. The resampling scheme proposed in (Dou and Song, 2024) can be used as is, to improve the quality of the samples from $p(\boldsymbol{\theta}|\boldsymbol{D}_{k-1})$ as this is a usual conditional distribution. The resampling scheme is based on the FPS update: $\boldsymbol{\theta}_j^{(t)}$ is first moved using the backward SDE into $\boldsymbol{\theta}_j^{(t+1)}$ according to $p(\boldsymbol{\theta}^{(t+1)}|\boldsymbol{\theta}^{(t)},\boldsymbol{y}^{(T-t-1)},\boldsymbol{\xi},\boldsymbol{D}_{k-1})$, which satisfies

$$p(\boldsymbol{\theta}^{(t+1)}|\boldsymbol{\theta}^{(t)},\boldsymbol{y}^{(T-t-1)},\boldsymbol{\xi},\boldsymbol{D}_{k-1}) \propto p_t(\boldsymbol{\theta}^{(t+1)}|\boldsymbol{\theta}^{(t)},\boldsymbol{D}_{k-1})\, p(\boldsymbol{y}^{(T-t-1)}|\boldsymbol{\theta}^{(t+1)},\boldsymbol{\xi}) \quad (53)$$

where $p_t(\boldsymbol{\theta}^{(t+1)}|\boldsymbol{\theta}^{(t)},\boldsymbol{D}_{k-1})$ is given in closed form by the unconditional diffusion model and $p(\boldsymbol{y}^{(T-t-1)}|\boldsymbol{\theta}^{(t+1)},\boldsymbol{\xi})$ is given by (47). As both these distributions are Gaussian, $p(\boldsymbol{\theta}^{(t+1)}|\boldsymbol{\theta}^{(t)},\boldsymbol{y}^{(T-t-1)},\boldsymbol{\xi},\boldsymbol{D}_{k-1})$ can be written in closed form and resampling weights can be written as:

$$w_i = \frac{\tilde{w}_i}{\sum_{i=1}^{N} \tilde{w}_i} \quad \text{with} \quad \tilde{w}_i = p(\boldsymbol{y}^{(T-t-1)}|\boldsymbol{\theta}_i^{(t)},\boldsymbol{\xi}) \quad (54)$$

where $p(\boldsymbol{y}^{(T-t-1)}|\boldsymbol{\theta}_i^{(t)},\boldsymbol{\xi})$ is tractable (see Dou and Song (2024) for more details).

For the pooled posterior $q_{\boldsymbol{\xi},N}(\boldsymbol{\theta}') \propto \prod_{i=1}^{N} p(\boldsymbol{\theta}'|\boldsymbol{y}_i,\boldsymbol{\xi},\boldsymbol{D}_{k-1})^{\nu_i}$, update (53) takes the form:

$$\prod_{i=1}^{N} p(\boldsymbol{\theta}^{(t+1)}|\boldsymbol{\theta}^{(t)},\boldsymbol{y}_i^{(T-t-1)},\boldsymbol{\xi},\boldsymbol{D}_{k-1})^{\nu_i} \propto p_t(\boldsymbol{\theta}^{(t+1)}|\boldsymbol{\theta}^{(t)},\boldsymbol{D}_{k-1}) \prod_{i=1}^{N} p(\boldsymbol{y}_i^{(T-t-1)}|\boldsymbol{\theta}^{(t+1)},\boldsymbol{\xi})^{\nu_i}$$

$$\propto \prod_{i=1}^{N} \left( p(\boldsymbol{\theta}^{(t+1)}|\boldsymbol{\theta}^{(t)},\boldsymbol{D}_{k-1})\, p(\boldsymbol{y}_i^{(T-t-1)}|\boldsymbol{\theta}^{(t+1)},\boldsymbol{\xi}) \right)^{\nu_i}$$

$$(55)$$

which leads to the following resampling weights:

$$w'_j = \frac{\tilde{w'_j}}{\sum_{j=1}^{M} \tilde{w'_j}} \quad \text{with} \quad \tilde{w'_j} = \prod_{i=1}^{N} p(\boldsymbol{y}_i^{(T-t-1)}|\boldsymbol{\theta}_j'^{(t)},\boldsymbol{\xi})^{\nu_i} \ .$$

# F  Numerical experiments

## F.1  Sequential prior contrastive estimation (SPCE) and Sequential nested Monte Carlo (SNMC) criteria

The SPCE introduced by Foster et al. (2021) is a tractable quantity to assess the design sequence quality. For a number $K$ of experiments, $\boldsymbol{D}_K = \{(\boldsymbol{y}_1,\boldsymbol{\xi}_1),\cdot,(\boldsymbol{y}_K,\boldsymbol{\xi}_K)\}$ and $L$ contrastive variables, SPCE is defined as

$$SPCE(\boldsymbol{\xi}_1,\cdot,\boldsymbol{\xi}_K) = \mathbb{E}_{\substack{K \\ \prod_{k=1}^{K} p(\boldsymbol{y}_k|\boldsymbol{\xi}_k,\boldsymbol{\theta}_0) \ \prod_{\ell=0}^{L} p(\boldsymbol{\theta}_\ell)}} \left[ \log \frac{\prod_{k=1}^{K} p(\boldsymbol{Y}_k|\boldsymbol{\theta}_0,\boldsymbol{\xi}_k)}{\frac{1}{L+1} \sum_{\ell=0}^{L} \prod_{k=1}^{K} p(\boldsymbol{Y}_k|\boldsymbol{\theta}_\ell,\boldsymbol{\xi}_k)} \right] . \quad (56)$$

SPCE is a lower bound of the total EIG which is the expected information gained from the entire sequence of design parameters $\boldsymbol{\xi}_1, \ldots, \boldsymbol{\xi}_K$ and it becomes tight when $L$ tends to $\infty$. In addition, SPCE has the advantage to use only samples from the prior $p(\boldsymbol{\theta})$ and not from the successive posterior distributions. It makes it a fair criterion to compare methods on design sequences only. Considering a true parameter value denoted by $\boldsymbol{\theta}^*$, given a sequence of design values $\{\boldsymbol{\xi}_k\}_{k=1:K}$, observations $\{\boldsymbol{y}_k\}_{k=1:K}$ are simulated using $p(\boldsymbol{y}|\boldsymbol{\theta}^*, \boldsymbol{\xi}_k)$ respectively. Therefore, for a given $\boldsymbol{D}_k$, the corresponding SPCE is estimated numerically by sampling $\boldsymbol{\theta}_1, \cdot, \boldsymbol{\theta}_L$ from the prior,

$$SPCE(\boldsymbol{D}_K) = \frac{1}{N} \sum_{i=1}^{N} \left\{ \log \frac{\prod\limits_{k=1}^{K} p(\boldsymbol{y}_k|\boldsymbol{\theta}^*, \boldsymbol{\xi}_k)}{\frac{1}{L+1} \left( \prod\limits_{k=1}^{K} p(\boldsymbol{y}_k|\boldsymbol{\theta}^*, \boldsymbol{\xi}_k) + \sum\limits_{\ell=1}^{L} \prod\limits_{k=1}^{K} p(\boldsymbol{y}_k|\boldsymbol{\theta}_\ell^i, \boldsymbol{\xi}_k) \right)} \right\} .$$

Similarly, an upper bound on the total EIG has also been introduced by Foster et al. (2021) and named the Sequential nested Monte Carlo (SNMC) criterion,

$$SNMC(\boldsymbol{\xi}_1, \cdot, \boldsymbol{\xi}_K) = \mathbb{E}_{\substack{\prod\limits_{k=1}^{K} p(\boldsymbol{y}_k|\boldsymbol{\xi}_k, \boldsymbol{\theta}_0) \prod\limits_{\ell=0}^{L} p(\boldsymbol{\theta}_\ell)}} \left[ \log \frac{\prod\limits_{k=1}^{K} p(\boldsymbol{Y}_k|\boldsymbol{\theta}_0, \boldsymbol{\xi}_k)}{\frac{1}{L} \sum\limits_{\ell=1}^{L} \prod\limits_{k=1}^{K} p(\boldsymbol{Y}_k|\boldsymbol{\theta}_\ell, \boldsymbol{\xi}_k)} \right] .$$

As shown in Foster et al. (2021) (Appendix A), SPCE increases with $L$ to reach the total EIG when $L \to \infty$ at a rate $\mathcal{O}(L^{-1})$ of convergence. It is also shown in Foster et al. (2021) that for a given $L$, SPCE is bounded by $\log(L+1)$ while the upper bound SNMC below is potentially unbounded. As in Blau et al. (2022), if we use $L = 10^7$ to compute SPCE and SNMC, the bound is $\log(L+1) = 16.12$ for SPCE. In practice this does not impact the numerical methods comparison as the intervals [SPCE, SNMC] containing the total EIG remain clearly distinct.

## F.2 IMPLEMENTATION DETAILS

### F.2.1 SOURCE EXAMPLE

For VPCE (Foster et al., 2020) and RL-BOED (Blau et al., 2022), we used the code available at github.com/csiro-mlai/RL-BOED, using the settings recommended therein to reproduce the results in the respective papers. VPCE optimizes an EIG lower bound in a myopic manner estimating posterior distributions with variational approximations. RL-BOED is a non-myopic approach which does not provide posterior distributions. From the obtained sequences of observations and design values, we computed SPCE and SNMC as explained above and retrieved the same results as in their respective papers. For PASOA and SMC procedures, we used the code available at github.com/iolloj/pasoa. PASOA is a myopic approach, optimizing an EIG lower bound using sequential Monte Carlo (SMC) samplers and tempering to also provide posterior estimations. The method refered to as SMC is a variant without tempering.

For CoDiff, the $\nu_i$'s in the pooled posterior distribution were set to $\nu_i = \frac{1}{N}$. The current prior and posterior distributions at experimental step $k$ were initialized using respectively the prior and posterior samples at step $k - 1$. Design optimization was performed using the Adam optimizer with an exponential learning rate decay schedule with initial learning rate $10^{-2}$ and decay rate $0.98$. The Langevin step-size in the DiGS method Chen et al. (2024) was set to $10^{-2}$. The joint optimization-sampling loop was run for 5000 steps. Figure 5 shows samples from the current prior $p(\boldsymbol{\theta}|\boldsymbol{D}_{k-1})$, which gradually concentrate around the true sources as $k$ increases. The additional Figure 6 shows, at some intermediate step $k$, samples from the current prior $p(\boldsymbol{\theta}|\boldsymbol{D}_{k-1})$ and from the pooled posterior distribution in comparison, to illustrate its contrastive nature.

### F.2.2 MNIST EXAMPLE

In this example, the likelihood easily derives from $\boldsymbol{Y} = \boldsymbol{A}_{\boldsymbol{\xi}}\boldsymbol{\theta} + \boldsymbol{\eta}$, where $\boldsymbol{Y}, \boldsymbol{\theta}$ and $\boldsymbol{\eta}$ are familiarly seen as arrays of pixels. The transformation $T_{\boldsymbol{\xi}, \boldsymbol{\theta}}$ is simply $T_{\boldsymbol{\xi}, \boldsymbol{\theta}}(\boldsymbol{U}) = \boldsymbol{A}_{\boldsymbol{\xi}}\boldsymbol{\theta} + \boldsymbol{U}$ with $\boldsymbol{U} = \boldsymbol{\eta}$ a Gaussian variable. However, to fit in our theoretical framework, images need to be treated as mappings over a continuous spatial domain, so that $\boldsymbol{\xi}$ can be seen as a continuous parameter and $\boldsymbol{A}_{\boldsymbol{\xi}}\boldsymbol{\theta}$ be differentiable with respect to $\boldsymbol{\xi}$. This is common in image processing and analysis, where

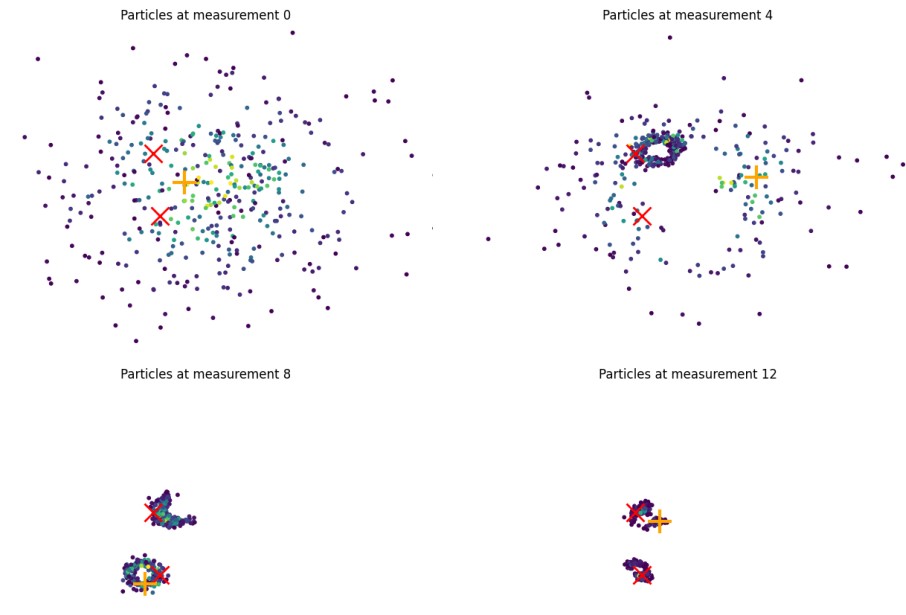

Figure 5: Source localization example. Experiments 0 (prior samples), 4, 8 and 12. As new design locations are selected (orange crosses), samples concentrate to the true sources (red crosses). Samples with lower weights in blue, higher weights in yellow.

2D images are often regarded as mappings over a continuous 2D domain (see *e.g.* Rein van den Boomgaard and Leo Dorst, 2021 Lecture Notes) with $x_1$ and $x_2$ axes. That is, for $(x_1, x_2) \in \mathbb{R}^2$, $\boldsymbol{\xi} = (\xi_1, \xi_2) \in \mathbb{R}^2$, we consider that an array of pixels $\boldsymbol{\theta}$ is a discrete sampled representation of a 2D function $\boldsymbol{\theta}(x_1, x_2)$ and more generally define, using abusively the same notation for continuous and sampled representations,

$$\boldsymbol{Y}(x_1, x_2) = \boldsymbol{A_\xi}(\boldsymbol{\theta}(x_1, x_2)) + \boldsymbol{\eta}(x_1, x_2)$$

where $\boldsymbol{A_\xi}(\cdot)$ is a masking operator depending on some length $h$ and defined by

$$\begin{aligned} \boldsymbol{A_\xi}(\boldsymbol{\theta}(x_1, x_2)) &= \boldsymbol{\theta}(x_1, x_2) \quad \text{if } (x_1, x_2) \in S_{\boldsymbol{\xi}, h} \\ &= 0 \quad \text{otherwise} \end{aligned}$$

with $S_{\boldsymbol{\xi}, h} = \{(x_1, x_2) \in \mathbb{R}^2, \xi_1 - h\Delta_1 \le x_1 \le \xi_1 + h\Delta_1, \xi_2 - h\Delta_2 \le x_2 \le \xi_2 + h\Delta_2$, denoting by $\Delta_1$ and $\Delta_2$ the sampling distances along the $x_1$ and $x_2$ axes.

To be able to derive with respect to $\boldsymbol{\xi}$, we then need to consider a smooth version $\boldsymbol{\mu_{\xi, s}}$ of $\boldsymbol{A_\xi}$. This is classically done by convoluting with a 2D Gaussian kernel $G_{\boldsymbol{s}}$, *e.g.* the product of two 1D Gaussian kernels with positive scales $\boldsymbol{s} = (s_1, s_2)$. In practice, this consists in smoothing the sharp borders of the mask, noting that $\boldsymbol{\mu_{\xi, s}} \to \boldsymbol{A_\xi}$ when $\boldsymbol{s} \to \boldsymbol{0}$,

$$\begin{aligned} \boldsymbol{\mu_{\xi, s}}(x_1, x_2) &= (\boldsymbol{A_\xi}(\boldsymbol{\theta}) * G_{\boldsymbol{s}})(x_1, x_2) \\ &= (\boldsymbol{A_0}(\boldsymbol{\theta}) * G_{\boldsymbol{s}})(x_1 - \xi_1, x_2 - \xi_2) \\ &= \boldsymbol{\mu_{0, s}}(x_1 - \xi_1, x_2 - \xi_2) \end{aligned}$$

where the second equality uses that $\boldsymbol{A_\xi}(\boldsymbol{\theta}(x_1, x_2)) = \boldsymbol{A_0}(\boldsymbol{\theta}(x_1 - \xi_1, x_2 - \xi_2))$. It follows that, no matter what $\boldsymbol{A_0}(\boldsymbol{\theta})$ is, such a convolution with $G_{\boldsymbol{s}}$ makes $\boldsymbol{\mu_{0, s}}$ into a function that is continuous and

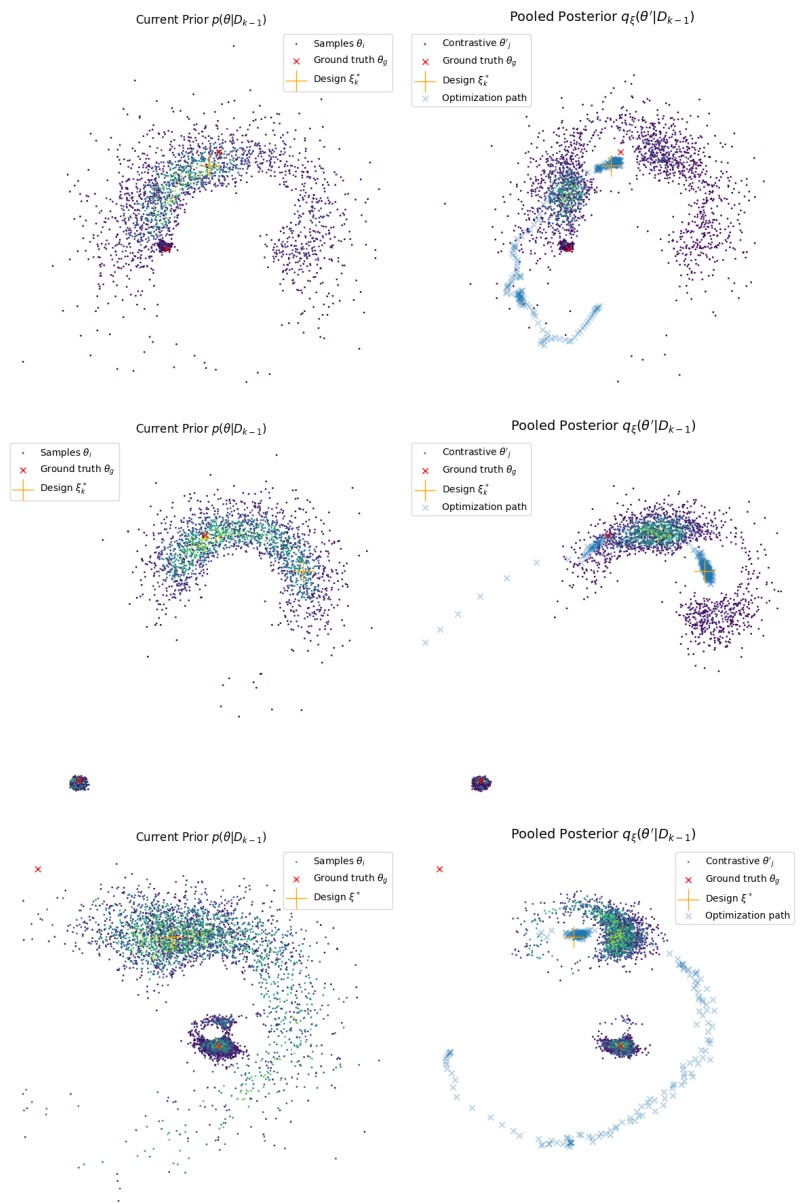

Figure 6: Several source localisation examples. Prior (left) and pooled posterior (right) samples at experiment $k$. Final $\boldsymbol{\xi}_k^*$ value (orange cross) at the end of the optimization sequence $\boldsymbol{\xi}_0, \cdot, \boldsymbol{\xi}_T$ (blue crosses). This optimization "contrasts" the two distributions by making the pooled posterior "as different as possible" from the prior.

infinitely differentiable in its arguments. Then, it comes, for $i = 1, 2$,

$$
\begin{aligned}
\frac{\partial \boldsymbol{\mu}_{\boldsymbol{\xi},s}}{\partial \xi_i}(x_1, x_2) &= \frac{\partial \boldsymbol{\mu}_{\boldsymbol{0},s}}{\partial \xi_i}(x_1 - \xi_1, x_2 - \xi_2) \\
&= -\frac{\partial \boldsymbol{\mu}_{\boldsymbol{0},s}}{\partial x_i}(x_1 - \xi_1, x_2 - \xi_2) \\
&= -\left( \boldsymbol{A_0}(\boldsymbol{\theta}) * \frac{\partial G_{\boldsymbol{s}}}{\partial x_i} \right)(x_1 - \xi_1, x_2 - \xi_2).
\end{aligned}
$$

These latter derivatives are all that is needed to define the gradients in our developments. If $\boldsymbol{A}_0$ is a Heaviside step function, its convolution with a distribution leads to the cumulative density function (cdf) of this distribution. The same developments are valid by replacing the Gaussian kernel by a bivariate logistic distribution $L$ with mean $\boldsymbol{0}$ and scales $\boldsymbol{s} = (s_1, s_2)$. The multivariate logistic distribution (Gumbel, 1961; Malik and Abraham, 1973) generalizes the univariate logistic distribution. Its pdf is $L_2(x_1, x_2; \boldsymbol{s}) = \frac{2! \exp(-x_1/s_1 - x_2/s_2)}{s_1 s_2 (1 + \exp(-x_1/s_1) + \exp(-x_2/s_2))^3}$. Its cdf is closed-form and is $\frac{1}{1 + \exp(-x_1/s_1) + \exp(-x_2/s_2)}$. In practice, we simply consider a product of two independent univariate logistic distributions with mean 0 and scale $s_i$, $i = 1, 2$, $L_1(x_i; s_i) = \frac{\exp(-x_i/s_i)}{s_i(1 + \exp(-x_i/s_i))^2}$. The cdf of a 1D logistic distribution is the sigmoid function $S(x_i; s_i) = \frac{1}{1 + \exp(-x_i/s_i)}$. The convolution of a Heaviside step function with such a logistic distribution is thus a smooth sigmoid. In the MNIST example, the mask length is set to $h = 7$, with sampling distances $\Delta_1 = \Delta_2 = 1$ and we use a 2D product logistic kernel with $s_1 = s_2 = 0.1$. Using that the 2D mask $\boldsymbol{A}_{\boldsymbol{\xi}}$ can be written as the following product $(H(x_1 - \xi_1 + h) + H(\xi_1 + h - x_1) - 1)(H(x_2 - \xi_2 + h) + H(\xi_2 + h - x_2) - 1)$, where $H$ is the 1D Heaviside step function, it follows that the smooth $\boldsymbol{\mu}_{\boldsymbol{\xi},s}$ is

$$
\boldsymbol{\mu}_{\boldsymbol{\xi},s}(x_1, x_2) = (S(x_1 - \xi_1 + h; s_1) + S(\xi_1 + h - x_1; s_1) - 1)(S(x_2 - \xi_2 + h; s_2) + S(\xi_2 + h - x_2; s_2) - 1).
$$

For the numerical example of Section 6.3, we used the MNIST dataset (LeCun et al., 1998), the time varying SDE (34) with a noise schedule $\beta(t) = b_{min} + (b_{min} - b_{max})(t - t_0)/(T - t_0)$ (with $b_{max} = 5$, $b_{min} = 0.2$, $t_0 = 0$, $T = 2$). The training of the usual score matching was done for 3000 epochs with a batch size of 256 and using Adam optimizer Kingma and Ba (2015). We used gradient clipping and the training was done on a single A100 GPU.

Update equations for the sampling operators were derived from SDE (19) for the contrastive samples of the pooled posterior $q_{\boldsymbol{\xi},N}(\boldsymbol{\theta}')$ and (45) for samples from the current prior $p(\boldsymbol{\theta}|\boldsymbol{D}_{k-1})$, where $\boldsymbol{D}_{k-1}$ can be added in the conditioning part without difficulty. Those updates are equivalent to (55) and (53) respectively. The resampling weights were computed as in Section E.

Figures 7 and 8 show additional image reconstruction processes. The digit to be recovered is shown in the first column. The successively selected masks are shown (red line squares) in the second column with the resulting gradually discovered part of the image. The reconstruction per se can be estimated from the posterior samples shown in the last 16 columns. At each experiment, the upper sub-row shows the 16 most-likely reconstructed images, while the lower sub-row shows the 16 less-probable ones. As the number of experiments increases the posterior samples gradually concentrate on the right digit.

Figure 9 then shows that design optimization is effective by showing better outcomes when masks locations are optimized (second column) than when masks are selected at random centers (third column). The highest posterior weight samples in the last 14 columns also clearly show more resemblance with the true digit in the optimized case. The superior performance of design optimization is confirmed quantitatively in Table 1, which reports the reconstruction quality as measured by the structural similarity index measure (SSIM) (Wang et al., 2004), for both CoDiff and random design. 20 ground truth digit images are randomly selected and the SSIM is computed for the CoDiff and random reconstructions, after each successive experiment out of 6. Table 1 reports the median SSIM over the 20 selected digits. The SSIM is a decimal value between -1 and 1, where 1 indicates perfect similarity, 0 indicates no similarity, and -1 indicates perfect anti-correlation.

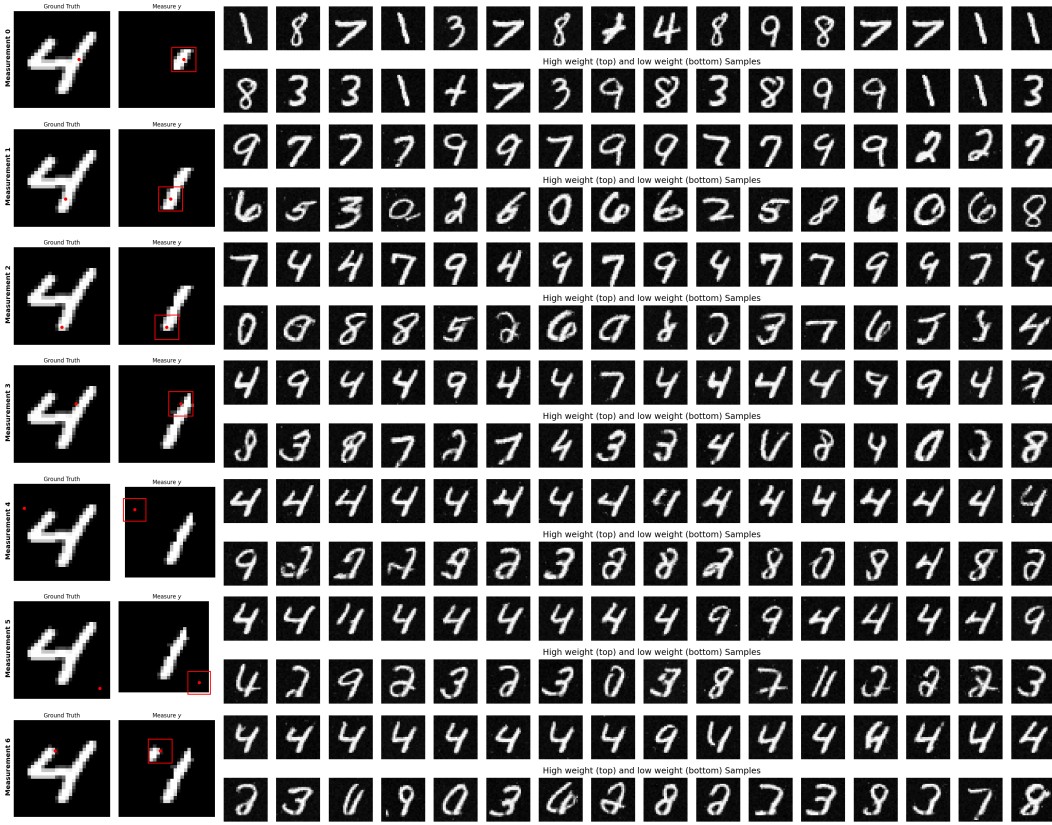

Figure 7: Image reconstruction. First 7 experiments (rows): image ground truth, measurement at experiment $k$, samples from current prior $p(\boldsymbol{\theta}|\boldsymbol{D}_{k-1})$, with best (resp. worst) weights in upper (resp. lower) sub-row. The samples incorporate past measurement information as the procedure advances.

## F.3 HARDWARE DETAILS

The source example 6.2 can be run locally. It was tested on an Apple M1 Pro 16Gb chip but faster running times can be achieved on GPU. The MNIST example 6.3 was run on a single A100 80Gb GPU.

## F.4 SOFTWARE DETAILS

Our code is implemented in Jax Bradbury et al. (2020) and uses Flax as a Neural Network library and Optax as optimization one Babuschkin et al. (2020). The code is available at https://github.com/jcopo/ContrastiveDiffusions.

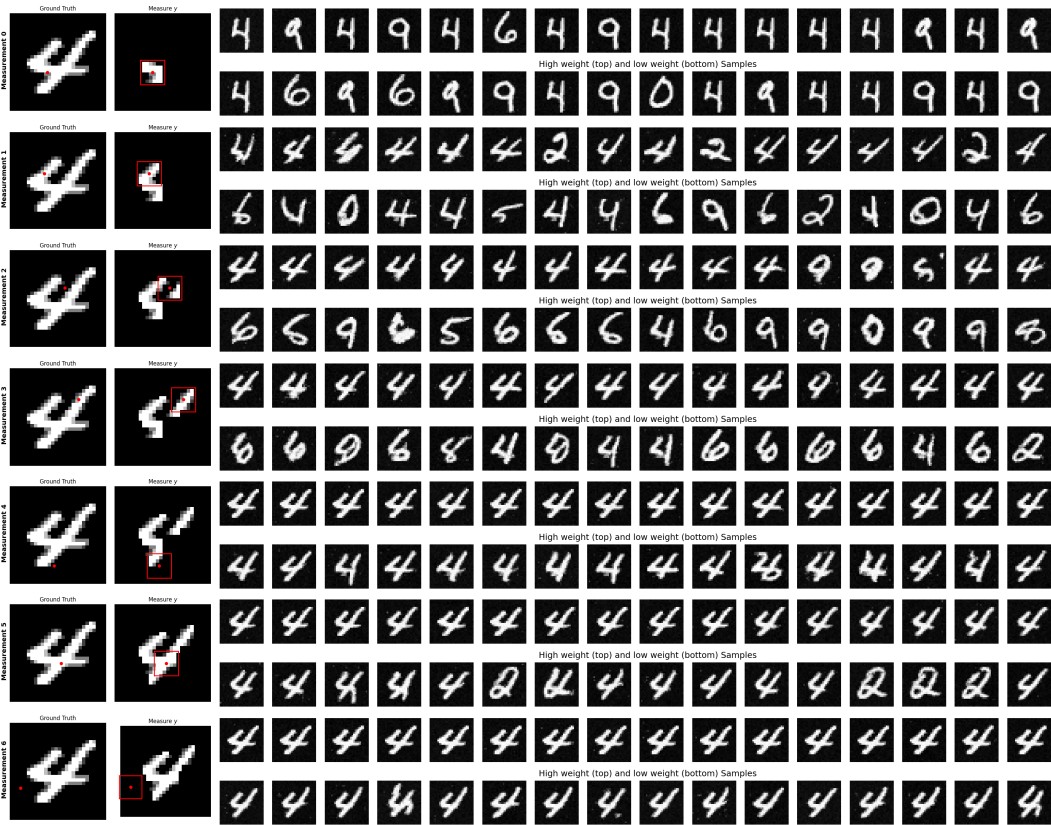

Figure 8: Image reconstruction. First 7 experiments (rows): image ground truth, measurement at experiment $k$, samples from current prior $p(\boldsymbol{\theta}|\boldsymbol{D}_{k-1})$, with best (resp. worst) weights in upper (resp. lower) sub-row. The samples incorporate past measurement information as the procedure advances.

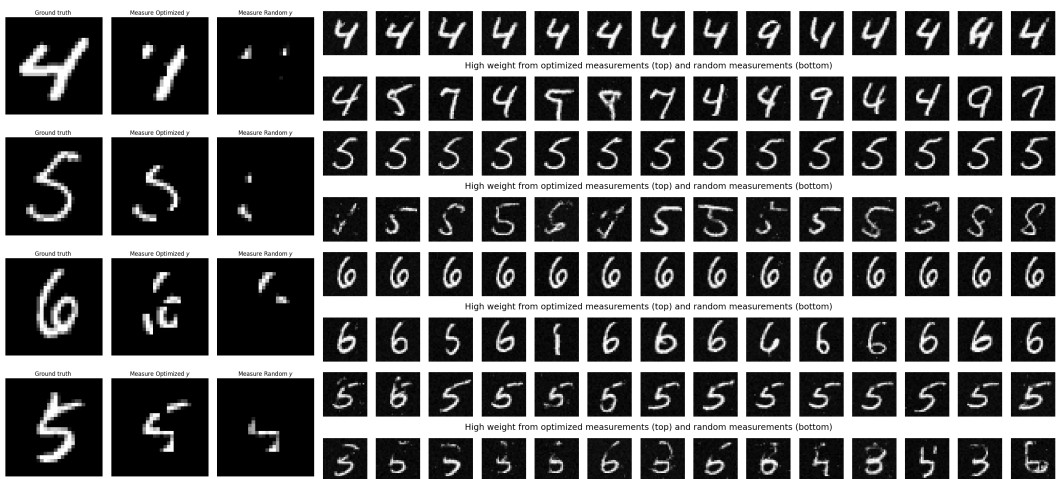

Figure 9: Image $\boldsymbol{\theta}$ (1st column) reconstruction from 7 sub-images $\boldsymbol{y} = \boldsymbol{A}_{\boldsymbol{\xi}}\boldsymbol{\theta} + \boldsymbol{\eta}$ selected sequentially at 7 central pixel $\boldsymbol{\xi}$. Optimized vs. random designs: measured outcome $\boldsymbol{y}$ (2nd vs. 3rd column) and parameter $\boldsymbol{\theta}$ estimates (reconstruction) with highest weights (upper vs. lower sub-row).

