# OpenReview forum: "Bayesian Experimental Design Via Contrastive Diffusions"
_ICLR.cc/2025/Conference — ICLR 2025 Spotlight_

### Official Review · Reviewer_VEZf · 2024-10-29

**Soundness:** 3
**Presentation:** 3
**Contribution:** 3
**Rating:** 8
**Confidence:** 4

**Summary:**

This paper makes an interesting connection between Bayesian optimal experimental design (BOED) and optimization by sampling using a diffusion model. While an interesting connection that may prove fruitful with some refinement, the authors seem to misunderstand the basis of BOED and maximization of a lower bound of mutual information using contrastive learning, as well as miss prior work that might better-inform this method. This disconnect, as well as confusing notation regrading how the diffusion model is used in multi-round BOED and lackluster empirical evaluation, makes me think this work is too early and could benefit from refinement in writing and additional experimental evaluations of the method.

**Strengths:**

- The authors make a great connection to optimization by sampling via diffusion models.
- Thorough gradient analysis.
- The connection to inverse problems e.g. inpainting in MNIST images as a BOED problem is an interesting new task in the BOED setting and could see BOED objectives (really mutual information objectives) gain further adoption in generative modeling.

**Weaknesses:**

While the authors make a great connection to optimization by sampling, theoretical analysis of the convergence of the desired goal distribution is missing. This would e.g. especially be helpful in the multi-step BOED setting to show how to improve bounds of the mutual information. I list out my critiques and questions below:

- The authors miss that they are also optimizing a lower bound of the mutual information by the InfoNCE bound, thus their comments that their approximation is exact is incorrect.
- The definition of information in Equation 2 does not make sense. The KL divergence is written in a non-standard way - do you mean $E_{p(y|\xi)}KL(p(\theta|y, \xi) || p(\theta))$?
- The authors miss the connection to Fotster et al. 2019b of LF-ACE that uses a posterior estimate in the InfoNCE bound to refine samples. e.g. in equation 5.
- The notation for the expected posterior in equation 9, it's not clear what is the proposal distribution $q(\theta)$ in the paper. Are you using likelihood evaluations? Also, how are you choosing the weighting of $\nu_i$?
- I would prefer to see the notation for iterative sampling operators similar to Marion et al. 2024, where the Y, and $\Theta$ are variables of the operator instead of in the superscript. I think that would make the optimization problem more clear.
- After equation 15, are you using two generative models to define the observed distribution $y$ and the parameter distribution $\theta$?
- in terms of the location finding experiment, usually we would expect to see the designs and final posterior distribution. the way it's portrayed in the paper is confusing. I'm not sure which samples are shown, which design round the designs come from, and why the seeming optimal design isn't on the ground truth values. It looks like the chosen design is quite far from the true theta_0.
- While the EIG seems to improve over others in Figure 3, the upper bound is _much_ worse. Why is that?
- The MNIST experiment is interesting but lacks quantitative evaluation. Maybe including a classifier and using the number of successfully classified images would help? Also, there seems to be opportunity to change the design size to see how that influences outcomes.

In summary, I think the notation is a little bloated. The authors tend to repeat themselves in the text and could be more concise. Concision would help make space for exposition of more salient points that would improve this paper. Additionally, it would help the authors to explicitly lay out what are the likelihood, posterior, and priors they use in experiments. Also, it would be nice to see calibration of the posterior that results of the final design round using e.g. the l-C2ST method or other simulation based calibration methods. Most applications of amortized posteriors using BOED will be interested in this, but it can be computationally expensive, which is why I think just on the final round is sufficient. This is an interesting method that I think needs cleaner exposition.

**Questions:**

My questions are contained in the Weaknesses section.

---

> ### Author Response · Authors · 2024-11-13
> **Clarification request about prior work**
>
> We thank the reviewer for this review and for acknowledging the "interesting contribution" and "great connection" made by our paper.
> We are of course willing to clarify all points raised, but we are currently facing some difficulties in understanding some of the reviewer critiques. The reviewer claims we have "misunderstood the basis of BOED" and refer to a reference "Fotster et al 2019b" and to LF-ACE. We suspect a misspelling of "Foster et al 2019b", but such reference does not exist in our bibliography. Does LF-ACE instead refer to "Likelihood-free ACE" in section 3.4 of "Foster et al 2020- A Unified Stochastic Gradient Approach to Designing Bayesian-Optimal Experiments" ?
>
>
> LF-ACE, $I_{ACE}$, and InfoNCE have the same structure that leads to gradient estimations different from ours. Taking $I_{ACE}$ from eq. (11) of Foster et al 2020:
>
> $$
> I_{\text{ACE}}(\xi, \phi, L)
> = E_{p(y | \theta_0, \xi) p(\theta_0)} \left[ \log p(y | \theta_0, \xi) - \log \left(\frac{1}{L+1} \sum_{\ell=0}^{L} \frac{p(\theta_\ell)}{q_\phi(\theta_\ell | y)} p(y | \theta_\ell, \xi)\right) \right]
> $$
>
> As visible in eq. (19) of Foster et al 2020, its gradient $\nabla_\xi I_{ACE}$ differs from our gradient form:
>
> $$
> \nabla_\xi EIG(\xi) = E_{p(y|\theta_0,\xi)p(\theta_0)} \left[ \nabla_\xi \log p(y | \theta_0, \xi) - E_{q(\theta' | y, \xi)} \left( \frac{p(\theta' | y, \xi)}{q(\theta' | y, \xi)} \nabla_\xi \log p(y | \theta', \xi) \right) \right]
> $$
>
>
> $I_{ACE}$ is both a lower bound of the EIG and  based on a variational approximation of the posterior, which is orthogonal to our approach. Also, we do not use a variational approximation of the posterior and instead use diffusion based samplers.
>
> For this reason, we believe the statement from the reviewer that our method "optimizes a lower bound of the mutual information by the InfoNCE bound" is not correct. Also, nowhere in the paper we write that our "approximation is exact". We focus mainly on the EIG gradient and not so much on the EIG itself. We do not emphasize the nature of the EIG approximation, which could be a lower bound, an upper bound or not a bound at all.
> Having said that we might have  "missed prior work that might better-inform our method" as the reviewer suggests. In such case, we are open to discussion but would appreciate specific references and detailed insights in what we missed.

---

> > ### Comment · Reviewer_VEZf · 2024-11-14
> > **Follow up to clarification about prior work**
> >
> > I thank the authors for answering some of my questions so quickly. Indeed, I was referring to LF-ACE from "A Unified Stochastic Gradient Approach to Designing Bayesian-Optimal Experiments". I appreciate the authors taking the time to point out the gradient is different but they still miss a core of my question. If this isn't a lower bound on the information or an approximation, what is the EIG they are optimizing? I'm not talking about the gradient of the estimator but the estimator itself. Indeed, if it's not a lower bound by the InfoNCE using the method from Foster et al 2020, then what is it?
> >
> > As for my comment about a direct approximation of the EIG, the authors are correct that they do not state that explicitly in their work and I was wrong. I assumed it by extension from my previous question and the allusion in line 114. I see that the authors are relying heavily on the work of Goda et al. 2022 and Ao and Li, 2024. I'll note they also do not explicitly state which bound of the mutual information they use. Following equation 2 from that paper,
> > $U(\lambda) = \mathbb{E}_{p(\theta|y, \lambda)}\left[\log l(y|\theta, \lambda)\right] - \mathbb{E}_{p(y|\lambda)}\left[\log p(y|\lambda)\right],$
> >  I can assume it's the InfoNCE lower bound since this the log ratio of the likelihood to marginal likeliihood. By extension, it can be shown the ACE gradient in equation 16 of Foster et al. 2020 is equivalent to the estimator in this paper since the L contrastive samples are used in a similar manner as the contrastive samples from the expected posterior in this paper. I'm dwelling on this because we know you can improve the information gain by increasing L but I'm encouraging the authors to think about how the expressiveness of the score function that approximates the expected posterior can be improved either by choice of importance sampling or neural network.
> >
> > I'm also still wondering why the SNMC upper bound for their method is much worse than the others. It honestly doesn't matter in practice but it's a weird result to display.
> >
> > I have more questions and comments after thinking this through and thanks to the authors clarification:
> > - What is the architecture of the diffusion model that's central to this paper? There's no mention and no code to reproduce the results. Appendix F has some details but I think more is needed to reproduce the results.
> > - How is the diffusion model pretrained since it's not being optimized in Algorithm 2?
> > - In line 365 the authors claim that running a conditional diffusion model is intractable yet that's exactly what this [paper](https://arxiv.org/abs/2209.14249) does so I don't think that method can be ruled out.
> > - I agree with the other reviewers you should state for the BOED community that you rely on explicit likelihoods.
> >
> > I apologize to the authors for being so pedenatic. I find this paper interesting and would like to understand it better with their help in explanations. Right now I'm left confused by the organization and some missing pieces that I think we can resolve.

---

> > > ### Author Response · Authors · 2024-11-14
> > > **Follow up #1**
> > >
> > > Thank you for your reactivity. We appreciate your acknowledging your assumption was incorrect. Your response raises interesting questions but we feel several statements made are ill-informed and some are inaccurate. We try to provide further clarifications and point out where we disagree below. To make clearer to which of your comments we reply, we put them in bold characters.
> > >
> > > ## Clarifications
> > > -  **"If this isn't a lower bound on the information or an approximation, what is the EIG they are optimizing? Indeed, if it's not a lower bound by the InfoNCE using the method from Foster et al 2020, then what is it?"**
> > >
> > > We do understand the core of your question and we feel it is an interesting question: You are asking what is the function we are optimizing with our new EIG gradient estimator and if by any chance this function is a lower bound or is linked to an existing lower bound?
> > > To make it short first, the answer is actually in the question: since we start from an EIG gradient, the function we optimize is the EIG! But the detailed answer is longer, let's explain better.
> > > As explained in our related work section lines 114 to 120, our approach belongs to approaches that **do not** start from an EIG lower bound to compute then the gradient of this lower bound. Instead, we try to handle directly the EIG gradient. We compute an exact (but intractable) expression of the gradient and then try to estimate it. So doing, the function we optimize is ideally the EIG itself and speaking about a bound is not useful anymore.
> > > **However**, the reviewer question is interesting because in practice of course the EIG gradient has to be approximated, for instance with our equation (10). Then it would be interesting to express the right-hand-side of (10) as the gradient of something. This something would be the "estimator" the reviewer is asking for. Unfortunately, it is not easy to express (10) as a gradient over $\xi$ because $w_{ij}$ depends on $\xi$ in a non simple manner. So to answer the question about what is this estimator we are optimizing, we are not aware of an explicit simple expression of this estimator that could be useful or interpreted as a bound or not. But **note that the whole point of our approach (like that of Goda et al and Ao and Li) is that we target the EIG itself and not a bound**. We are not saying that this is necessary always better but the spirit is different. Also, another possible justification of this direct approach is that, in gradient-based BOED, the EIG value is not needed, we just need to maximize it.
> > >
> > >
> > >
> > > - **"In line 365 the authors claim that running a conditional diffusion model is intractable yet that's exactly what this paper does so I don't think that method can be ruled out."**
> > >
> > > As explained in Appendix C.2, running a conditional diffusion model without retraining the score function is intractable as we cannot directly evaluate $\nabla_{\theta} \log p_t(y|\theta^{(t)}, \xi)$. There are many recent works tackling this problem and we refer to Daras et al., 2024 for a recent extensive review. While we appreciate the reference you provided, we find that it addresses relatively simple synthetic examples that may not generalize well to the MNIST outpainting tasks that we are tackling.
> > >
> > > - **"you should state for the BOED community that you rely on explicit likelihoods"**
> > >
> > > Please note that we do state this on lines 41, 219, 360, 432, 709 and 1000.
> > > In line 536 we then mention, as future work, possible extensions of our method, with no explicit expression of the likelihood. We can point this as a current limitation of our method.

---

> > > > ### Author Response · Authors · 2024-11-14
> > > > **Follow up #2**
> > > >
> > > > ## What we disagree on
> > > > - **"it can be shown the ACE gradient in equation 16 of Foster et al. 2020 is equivalent to the estimator in this paper since the L contrastive samples are used in a similar manner as the contrastive samples from the expected posterior in this paper"**
> > > >
> > > > This is not the case. The $I_{ACE}$ gradient eq (16) in Foster et al 2020 is different from what we propose. Note that the symbol $g$ is used for different things in (16) and in our eq (7).
> > > >  Our estimator and an estimator of equation (16) of Foster et al. 2020 target different quantities. Ours targets the gradient of the EIG and theirs targets the gradient of a lower bound of the EIG. They have no reason to be equivalent. The L contrastive samples are not used in a similar manner as our samples from the expected posterior. They are not drawn from the same distribution and contribute differently to the two estimators:
> > > > Foster has the form of a log of a sum: $-\log \left(\frac{1}{L+1} \sum_{\ell=0}^{L} \frac{p(\theta_\ell)}{q_\phi(\theta_\ell | y_i)} p(y_i | \theta_\ell, \xi)\right)$ while ours is:  $-\frac{1}{M} \sum_{j=1}^{M} w_{i, j}\; \nabla_\xi \log p(y_i | \theta_j, \xi)$
> > > >
> > > >
> > > > - **"SNMC upper bound for their method is much worse than the others"**
> > > >
> > > > The reviewer's claim is not correct. The SNMC upper bound of CoDiff is above all the other methods. As the true EIG is likely to keep increasing with the number of experiments, it is usually considered as a good feature that the upper bound is not itself bounded or does not plateau. See for instance a similar behavior in Tables 1 and 3 of Foster et al 2021 or in Table 1 of Blau et al 2022.
> > > >
> > > >
> > > >
> > > >
> > > > - **"no code to reproduce the results" and the "architecture of the diffusion model", "how is the diffusion model pretrained since it's not being optimized in Algorithm 2"**
> > > >
> > > > Please note that all the code is provided in the supplementary material. The architecture used is based of an Unet and is implemented in diffuse/unet.py. The diffusion model is pretrained in the same way as in Song et al. 2021. A brief overview is given in appendix C.1.
> > > >
> > > >
> > > >
> > > >
> > > > We hope we made the above points clearer. We also value and keep in mind your other suggestions, in particular on how to improve the readability of our paper. Synthetizing yours and the other reviewers comments, we will try to propose some improvements in our next answers.

---

> > > > > ### Comment · Reviewer_VEZf · 2024-11-17
> > > > > **follow up #3**
> > > > >
> > > > > Thank you for the detailed responses and clarifications. I apologize for not seeing the code earlier and thank the authors for their help.
> > > > >
> > > > > - I appreciate the authors pointing out what they mean by stating that the likelihood is available in closed form. I looked at their references and I see why I didn't catch it the first time. It would help to simply state "We assume throughout this paper that the prior and likelihood are available in closed form..." in the first sentence and maybe again when introducing the experiments would help readers. As it stands, this is hard to pick up on from a first read.
> > > > > - Let's move on from the lower bound debate. The EIG is defined with a term for the marginal likelihood in the denominator that relies on sampling to approach the true marginal likelihood as the number of samples approaches infinity. It's a lower bound in variational inference when you are optimizing a distribution to approximate the value. You're weighting the likelihood by an optimized expected posterior that is dependent on the $\xi$ parameter. As you optimize $\xi$ you return better probability estimates of the marginal likelihood, akin to variational inference. Also, I see you're using SPCE bound, which is a lower bound.
> > > > > - To determine whether SNMC is an upper bound or not, refer to Table 1 of [Blau et al. 2022](https://arxiv.org/pdf/2202.00821). You'll see they use it as an upper bound.
> > > > > - Why not report EIG for the MNIST experiment? That would help to see the "value" of performing each experiment. This is also related to [previous work](https://arxiv.org/pdf/1801.04062) on mutual information estimation on images that might be helpful to use as a baseline/comparison that I believe uses the NWJ MI bound, or to make more interesting connections.
> > > > > - I may be misunderstanding a key point, but why not compare to diffusion guidance to get a 'tilted' posterior? that could be a simpler process than using implicit diffusion to optimize experimental designs.
> > > > > - I would still like to see clearer figures for Figure 2 and more clear explanations.
> > > > >
> > > > > I thank the authors for their clarifications. The contribution of the paper is becoming more clear to me. If you have a pre-trained diffusion model of a dataset that you would like to use in BOED then you can optimize the designs using implicit diffusion. As it stands, I think the method is most interesting in the MNIST case. There are unanswered questions in the source finding problem as well as lack of clear explanation of the method. I agree with other reviewers that the paper seems hastily written and the technical exposition is hard to follow and seems stitched together at points. This is a big hurdle for interpretability. I think if the authors address these points it would improve the paper. I also think improved motivation of the MNIST problem, and problems related to it, would help expand the field of BOED to a more general ML audience.

---

> > > > > > ### Author Response · Authors · 2024-11-22
> > > > > > **Important Clarifications Needed**
> > > > > >
> > > > > > We appreciate the reviewer taking the time to fully understand our work. We still believe there are several important clarifications needed, which we outline below:
> > > > > >
> > > > > > - **About what the reviewer calls the lower bound debate**: We agree with the analysis made by the reviewer but this analysis applies to the EIG. **It does not apply to the EIG gradient.**
> > > > > > For instance the following sentence in your comment is correct but does not apply to our work:
> > > > > > "The EIG is defined with a term for the marginal likelihood in the denominator that relies on sampling to approach the true marginal likelihood"
> > > > > > The reason is doesn't apply is because we never compute the EIG to maximize it, we maximize it directly using its gradient. This sentence does not apply to our work because we do not start from an approximation of the EIG.
> > > > > > In short, previous methods do: 1) approximate the EIG by a lower bound, 2) Derive the **exact** gradient of this lower bound.
> > > > > > Instead we do: 1) Derive the **exact** gradient of the **exact** EIG, 2) approximate this gradient.
> > > > > > In other words, in previous cases, the approximation is made in step 1), while in our case the approximation is in 2). The two steps do not commute.
> > > > > > We explain the difference in the related work section lines 115 to 125. As we replied earlier, it's not obvious to us whether the two approaches can be linked, ie to see whether our approximate gradient is that of an EIG lower bound.
> > > > > >
> > > > > >
> > > > > >
> > > > > > - **Reference to Blau et al 2022**: we are happy that the reviewer acknowledge why we referred to their Table 1 in our previous reply entitled Follow up #2
> > > > > >
> > > > > >
> > > > > > - **The EIG is not shown for the MNIST example**: It is intractable and computing lower and upper bounds as in the source localization example would require 1e7 image samples which is infeasible. Instead we followed the reviewer recommandation of showing some more quantitative evaluation. We show now a Table reporting the reconstruction quality as measured by the SSIM index. As the task is an inpainting task this index is more relevant than a classification performance result.
> > > > > >
> > > > > > - **Diffusion guidance**: We are unsure of what the reviewer refers to here. Conditionnal sampling in diffusion can already be seen as guidance, see appendix C2.
> > > > > >
> > > > > > - **Clearer Figure 2 and source example**: For the source example, an additional figure 5 has been added in the Appendix to show that the posterior samples concentrate on the true location. In contrast, the fact that the design locations are not so close to the true sources is not necessary a problem, especially for the early experiments, as in the absence of much information, it is more interesting to explore the landscape. Figure 2 was included to show an intuition of the expected (now referred to as pooled) posterior.
> > > > > >
> > > > > >
> > > > > > - **Improving the paper presentation**: We understand from our interactions that the reviewer has trouble understanding our work and that results in critiques about the paper presentation. The presentation is certainly not perfect but it is misleading from the reviewer to include the other reviewers in the critique, as we note that the presentation is overall evaluated positively by the 2 other reviewers:
> > > > > >  j7MS: "the writing is clear and each idea is easy enough to follow".
> > > > > > 7k3H: "The method is presented clearly along with discussions w.r.t. to the related works. Overall, this is a well-written and technically solid paper".
> > > > > >
> > > > > >
> > > > > > We value all your comments that help us to improve the writting and we tried to provide an improved version, in particular for sections 4 and 5 as also mentioned by reviewer 7k3H, and related to prior works as mentioned by j7MS. See the summary of changes made in the new pdf.

---

> > > > > > > ### Comment · Reviewer_VEZf · 2024-11-24
> > > > > > > **Final clarifications**
> > > > > > >
> > > > > > > I really enjoyed the last edits and thing there are a few, critical, things to clarify before I can feel like this is ready for ICLR.
> > > > > > > - \textbf{Gradient of the EIG is not the same as the bounds} I'm still convinced you're optimizing a lower bound. The gradients are another point. If it helps the authors, an analogy is that it's like optimizing a conditional probability path in flow matching instead of the probability path itself. The gradients are proportional so it works out fine. I don't think this is a big issue and would like to see mention of "not using the lower bound" removed in the paper. Otherwise it's just false. There are just a few sentences and mentions.
> > > > > > > - \textbf{An upper bound should approach a lower bound} With reference to Blau, you'll notice that the goal of the upper bound is to "sandwich" the true MI with the lower bound. The closer the upper bound to the lower bound the better. What you're showing is that for some reason your approximation diverges in the upper bound. Like I said, I don't really know what that means but showing that figure raises more questions that I don't think should be the point of this paper - which is already doing enough to spread BOED to new applications with more general ML applications.
> > > > > > >
> > > > > > > These are the remaining few concerns I have before I can feel confident increasing the paper's score. I appreciate the authors' work to improve the interpretability of the paper as well.

---

> > > > > > > > ### Author Response · Authors · 2024-11-24
> > > > > > > > **Clarifications and Updated PDF**
> > > > > > > >
> > > > > > > > We thank the reviewer again for his time and efforts in reviewing this paper. We are glad our updated version was well received and are more than willing to address the last two points:
> > > > > > > >
> > > > > > > > - **for the reference to lower bounds:** we agree that there is one piece of sentence left in the paper that may be ambiguous. It is in the abstract line 20 *"without resorting to lower bound approximation of the EIG"* and we have now removed it in the updated pdf, as it is not important in the paper.
> > > > > > > > We also removed the last sentence at the end of the Related Work Section that might be a bit misleading too: "Additionally, they focus on lower-bound maximization which might sacrifice precision"
> > > > > > > > Besides those two locations, we don't think the issue is mentioned anywhere else in the paper in a way that might lead to wrong conclusions.
> > > > > > > >
> > > > > > > >
> > > > > > > > - **An upper bound should approach a lower bound (SPCE/SNMC)**: It can be shown that  $\text{SPCE} \leq log(L+1)$ (see Foster et al 2021, also recalled in our Appendix Lines 1075-1079) where $L$ is the number of constrastive samples used to compute SPCE and SNMC. For the sources, at the end of the design process most of the information has been captured and the lower bound SPCE saturates, meaning SPCE $\approx log(L+1)$. In contrast it is also shown in Foster et al 2021 (and mentioned in our appendix) that the upper bound SNMC is unbounded. Then, the "sandwich" SPCE/SNCM can get wider and wider. This SNMC increase is a good sign that the posterior concentrated to the ground truth positions of the sources and that the designs location are closer and closer to these ground truth positions.

---

> > > > > > > > > ### Author Response · Authors · 2024-11-27
> > > > > > > > >
> > > > > > > > > We thank the reviewer for updating the rating.
> > > > > > > > > We are grateful for your positive feedback and thank you for your interest and your commitment in engaging with us.

---

### Official Review · Reviewer_j7MS · 2024-11-04

**Soundness:** 3
**Presentation:** 2
**Contribution:** 3
**Rating:** 6
**Confidence:** 4

**Summary:**

This paper is a little outside of my expertise and I apologize in advance for any misunderstanding I may have and look forward to any correction from the authors. I have set my confidence to low accordingly.

The authors consider the problem of experimental design, given a statistical model of the form $P(y | \xi, \theta)$, we can choose values of $\xi$ and observe a corresponding $y$, and the goal is to infer $\theta$. In the first example in the paper, $\xi$ are locations on a 2D plane, $y$ are the measurements at the $\xi$ locations, and $\theta$ represents the source locations of contamination in the plane.

Assuming we have chosen the $\xi$ values, we can use the $p(y|\xi)$ to simulate $y_i$ values, one hallucinated rollout into the future, then infer the hallucinated posterior $p(\theta'|y_i, \xi)$ and measure the posterior distribution's entropy, $H(p(\theta'|y_i,\xi))$, typically requiring Monte-Carlo over the $\theta'$. We may repeat this process many many times for $i=1,...,N$, and compute the average posterior entropy, wrapping the inner Monte-Carlo over $\theta'$ in another outer Monte-Carlo over $y$. This average posterior entropy is the Mutual Information (MI) between $y$ and $\theta$ up to an additive constant (that is the prior entropy $H(p(\theta))$). This MI quantifies the benefit of sampling $y$ at $\xi$, we can therefore change $\xi$ in a way to maximise MI and this is a standard method for experimental design known as Expected Information Gain, or EIG.

However as evaluating EIG requires performing nested Monte-Carlo, and optimizing $\xi$ requires the gradient of EIG, much work has focused on various methods to construct efficient EIG gradient estimators. In past works and this works, the second inner integration over $\theta'$ is performed with MCMC or by Monte Carlo with importance sampling using a proposal $q(\theta'|y_i)$ that must be updated for every outer MC iteration $i$.

This work proposes to use a single proposal distribution across all outer MC iterations $i$ called the "expected posterior" which is a geometric mixture of all the hallucinated posteriors
$$
q(\theta') \propto \prod_i p(\theta|y_i, \xi)^{\nu_i}
$$

The paper proposes to Langevin dynamics to generate samples for the outer Monte Carlo $\\{y_i, \theta_i\\}$ from the joint distribution. As these samples are updated in a iterative manner, and the goal is to iteratively optimize $\xi$, these nested iterative procedures can be merged and updated together, if $\xi$ only incrementally moves, it is wasteful ti discard all the old samples $\\{y_i, \theta_i\\}$ but keep incrementing them so they're up-to-date.

Finally the proposed method is applied to two sequential design applications, the first is the source location example in which the $p(\theta)$, $p(y|\theta, \xi)$ have known analytical forms and the proposed method outperforms baselines. The second is where $p(\theta)$ cannot be evaluated but can be sampled, specifically a diffusion model trained on MNIST images, $\theta$ is a full image,  $\xi$ are pixel locations, and $y$ are the 7 X 7 patch of pixel values around $\xi$. The goal is to infer the full image by cherry-picking image patches.

**Strengths:**

- rigorously cites and discusses related ideas
- merging multiple nested loops into a single loop is intuitive idea and has been applied in multiple separate fields (training a VAE is effectively the EM algorithm was simplified into a single step, in Bayesian optimzation few-shot Knowledge Gradient)
- the writing is clear and each idea is easy enough to follow.

**Weaknesses:**

### Technical Comments
- __expected posterior__ the paper states that they aim to find one proposal distribution that covers all possible future posteriors and proposes a distribution that is a geometric mixing of hallucinated posteriors $q(\theta) \propto \prod_i p(\theta|y_i, \xi)^{\nu_i}$. The prior is the true expected posterior $p(\theta) = \mathbb{E}_Y[p(\theta|y)]$, exactly the average density of all possible posterior densities.  Why not use the prior $p(\theta)$ as the one global proposal distribution for integration over all the hallucinated posteriors? In the sequential case (introduced at the end of the paper)

### Writing Comments
- __Writing Density__ while the writing is thankfully clear, the paper cites and quickly introduces many ideas from prior works in quick succession making the paper very hard work to read for me, a lot of context switching making the paper very tiring. I would suggest reducing the details of referenced works, and providing higher level descriptions where possible and elaborating more on the novel parts of the proposed method.

**Questions:**

- Can the authors provide an intuition as to why the proposed expected posterior is preferable to the true expected posterior = prior? Is this purely for the sequential setting?
- how are the $\nu_i$ values determined in the expected posterior?
- what stop other methods from being used on the MNIST example? Is it just the computational cost from multiple nested loops and the sequential use case?

---

> ### Author Response · Authors · 2024-11-14
> **Expected posterior choice and other methods on MNIST**
>
> Thanks for your thorough overall positive evaluation and interesting comments.
>
> - **Expected posterior and $\nu_i$ choice**:
>
> Note first that in our paper, there is no such a thing as a "true" expected posterior but we realize that these terms can be misleading. "Expected posterior" is just the name we give to the geometric mixture we take as a proposal $q$ in eq (7) or (8). We justify this choice theoretically with lemma 2 in Appendix B.
>
> We understand then, that the reviewer means that the prior would be a good alternative choice for the proposal distribution $q$ in eq (7) or (8) because the prior can be written as $p(\theta) = E_{p(y|\xi)}[p(\theta|y, \xi)]$.
>
> Our choice of the geometric mixture instead is motivated by its optimality in terms of importance sampling. Our lemma 2 shows that the geometric mixture of the $p(\theta|y_i, \xi)$ is the distribution $q$ that minimizes the sum of the $KL(q,p(\theta|y_i, \xi))$. This property is not satisfied by the prior.
>
>
> More intuitively, the big difference between our expected posterior and the prior is the dependence of the former on $y_i$ and $\xi$. In particular, in our sequential BOED context, using the prior as a proposal in our gradient estimator would remove the dependence on the design $\xi$, which would remove the good properties explained lines 198-202.
>
>
>
> Value $\nu_i$ are set as $\frac1N$ but other choices are possible.
>
> - **Other methods on MNIST**:
>
> The main bottleneck of other methods is the number of needed $\theta$ samples to accurately optimize the EIG. For amortized methods like RL-BOED (Blau et al 2022 ) and DAD (Foster et al 2021) a high number of samples is needed to train the policy (eg 2000x2000 from Appendix D of DAD,  10 000 from Appendix C of RL-BOED).
> For images, you run out of memory on a GPU before 500 samples. So the training loops are already infeasible.
> On top of that, each time you need a new batch of samples you need to go through the whole denoising process again, which can become also time expensive. Both of the methods are practical when you have access to a $p(\theta)$ you can sample from efficiently.
> For Pasoa (Iollo et al. 2024) the number of samples needed for them to be efficient is higher than our CoDiff (see Fig. 3). There is also no straightforward way to do a usual Sequential Monte Carlo inference for images like MNIST. Their separate optimization/inference framework is also more time expensive than ours.

---

> ### Comment · Reviewer_j7MS · 2024-11-17
> **Thank you for the reply**
>
> Thank you for the thoughtful response.
>
> It seems neither of my comments are echoed by other reviewers reducing their weight somewhat.
>
> I will leave my score and confidence score as they are.

---

> > ### Author Response · Authors · 2024-11-22
> >
> > As you rightly pointed out, the previous name "expected posterior" to designate our new proposal distribution was misleading, as it evokes more naturally the prior than our proposal. We thus renamed our proposal distribution as the "pooled posterior" instead. See also other changes  made in a revised version of the paper listed in the summary.
> >
> > Thank you for the time and effort taken to review our paper.

---

### Official Review · Reviewer_7k3H · 2024-11-05

**Soundness:** 3
**Presentation:** 3
**Contribution:** 3
**Rating:** 8
**Confidence:** 2

**Summary:**

This paper presents a new expression for the gradient of the expected information gain (EIG)---a central quantity in Bayesian optimal experimental design (BOED). Their method, named CoDiff, combines the two steps of maximizing the EIG and then sampling from the posterior in BOED into a single joint step by leveraging the sampling-as-optimization setting of e.g. Marion et al. (2024). Doing so results in a more efficient BOED method, as demonstrated by numerical experiments.

**Strengths:**

The paper advances the field of BOED by making it more computationally efficient, whilst broadening the scope of problems that can be tackled by BOED by extending it to diffusion-based generative models. The idea of performing joint sampling and optimization using bi-level optimization seems novel to, although I am not very familiar with this literature. The experimental results are significant and demonstrate the efficacy of CoDiff compared to other baselines. The method is presented clearly along with discussions w.r.t. to the related works. Overall, this is a well-written and technically solid paper, and I am happy to recommend its acceptance.

**Weaknesses:**

Some minor points:
* It would be nice to see some uncertainty bars around the results in Figure 3.
* Parts of Section 4 and 5 were a bit difficult for me to follow, but that could just be because I am not familiar with the relevant literature. However, if possible, it might be a good exercise for the authors to think about making these parts a bit more accessible to the BOED community who are not familiar with the sampling-as-optimization literature.
* It would have been nice to empirically test the performance of CoDiff as e.g. the dimensions of $\theta$ or the hyperparameters (if any) vary.
* Please include some limitations of CoDiff in the conclusion section.

**Questions:**

See the "Weaknesses" section.

---

> ### Author Response · Authors · 2024-11-15
> **Reply to minor points**
>
> Thank you very much for your support and positive analysis. We are planning to prepare a new version of the paper where we try to improve the readability by working as suggested on Sections 4 and 5 and taking into account the other reviewers comments.
>
> Regarding your other minor points,
> -  Uncertainty bounds are actually plotted on Figure 3 but the figure needs to be zoomed to see them. We now added in the caption of Figure 3 that standard errors are also plotted. Their relatively low values is consistent with other similar results on this source location example. See for instance Table 1 in Blau et al 2022.
>
> - Regarding the dimension of $\theta$, the MNIST example already corresponds to a larger dimensional $\theta$ compared to what is found in the BOED literature, namely the dimension is 28*28=784.
> - In the revised version, we plan to reformulate the conclusion to make clearer the current limitations of CoDiff, namely that it remains a greedy approach, that an explicit expression of the likelihood is needed and that when using diffusion models, only linear forward models are currently handled.

---

### Author Response · Authors · 2024-11-22
**PDF revision summary**

The presentation in our paper is certainly not perfect but it was overall evaluated positively by 2 of the reviewers:
 j7MS: "the writing is clear and each idea is easy enough to follow".
7k3H: "The method is presented clearly along with discussions w.r.t. to the related works. Overall, this is a well-written and technically solid paper".

Nevertheless, as suggested by all 3 reviewers there is space for improvement. We thus provide a revised version of the paper with the following modifications, trying to account for reviewers' comments:

- Pooled posterior: as rightly pointed out by reviewer j7MS, the previous name "expected posterior" to designate our new proposal distribution was misleading, as it evokes more naturally the prior than our proposal. We thus renamed our proposal distribution as the "pooled posterior" instead.
- Partial rewritting, cleaning: as suggested by 7k3H, sections 4 and 5 have been simplified and made more concise as also suggested by VEZf. In particular, with less details on reference works and more highlevel descriptions (j7MS comment).
- Limitations have been included in the conclusion as suggested by 7k3H.
- The introduction has also been modified to  better motivate the diffusion/BOED combination (VEZf)
- About experiments: As suggested by VEZf, a quantitative evaluation has been added in the MNIST example, using the SSIM to evaluate reconstructions quality. The SSIM is a commonly used index in image reconstruction (https://en.wikipedia.org/wiki/Structural_similarity_index_measure)
- For the source example (VEZf), an additional figure has been added in the Appendix to show that the posterior samples concentrate on the true location. In contrast, the fact that the design locations are not so close to the true sources is not a problem, especially for the early experiments, as in the absence of much information, it is more interesting to explore the landscape.

---

### Meta-Review · Area_Chair_Zoa2 · 2024-12-16

**Metareview:**

The paper address the problem of Bayesian optimal experimental design (BOED), where the expected information gain EIG is the central quantity of interest. The paper presents CoDiff, a method that leverages the sampling-as-optimization approach to do the maximization of the EIG and sampling from the posterior. The paper improves the computational efficiency of the BOED and broadens its scope by incorporating diffusion models.   All the reviewers recommend acceptance and I agree.

**Additional Comments On Reviewer Discussion:**

All the reviewers recommend acceptance and I agree.

---

### Decision · Program_Chairs · 2025-01-22

Accept (Spotlight)